# The Polar Express: Optimal Matrix Sign Methods and Their Application to the Muon Algorithm

**Noah Amsel**
New York University
`noah.amsel@nyu.edu`

**David Persson**
New York University
Flatiron Institute
`dup210@nyu.edu`

**Christopher Musco**
New York University
`cmusco@nyu.edu`

**Robert M. Gower**
Flatiron Institute
`rgower@flatironinstitute.org`

## Abstract

Computing the polar decomposition and the related matrix sign function has been a well-studied problem in numerical analysis for decades. Recently, it has emerged as an important subroutine within the `Muon` optimizer for training deep neural networks. However, the requirements of this application differ sharply from classical settings: deep learning demands GPU-friendly algorithms that prioritize high throughput over high precision. We introduce `Polar Express`, a new method for computing the polar decomposition.[1] Like Newton-Schulz and other classical polynomial methods, our approach uses only matrix-matrix multiplications, making it very efficient on GPUs. Inspired by earlier work of Chen & Chow and Nakatsukasa & Freund, `Polar Express` adapts the update rule at each iteration by solving a minimax optimization problem. We prove that this strategy minimizes error in a worst-case sense, allowing `Polar Express` to converge as rapidly as possible both in the early iterations and asymptotically. We also address finite-precision issues, making it practical to use in `bfloat16`. When integrated into `Muon`, our method yields consistent improvements in validation loss for a GPT-2 model trained on one to ten billion tokens from the FineWeb dataset, outperforming recent alternatives across a range of learning rates.

## 1 Introduction

Advanced linear algebra is making its way into deep learning. Efficient algorithms for computing *matrix functions* have found exciting new applications in training neural networks. In particular, approximations to the matrix-inverse are used in the full Adagrad method (Duchi et al., 2011), the matrix square-root and quarter-root appear as subroutines in the Shampoo and Soap optimizers (Gupta et al., 2018; Shi et al., 2023; Vyas et al., 2025), and most recently, the matrix sign function has become a key ingredient of the `Muon` optimizer (Bernstein & Newhouse, 2024b;a; Jordan et al., 2024b). While the problem of computing these matrix functions has been studied by numerical analysts for decades, applications in deep learning come with different requirements than those in computational science. For deep learning, it is critical to take maximum advantage of GPU-friendly operations like matrix-matrix products and to avoid less parallel operations. Moreover, memory overhead must be small to handle large models. On the other hand, high accuracy is typically less important; the gold standard of sixteen digits of accuracy is overkill in deep learning.

Given these considerations, there is a need to develop new matrix function methods that are tailor-made for deep learning applications. We take on this challenge by designing a state-of-the-art, GPU-friendly algorithm for computing the matrix sign function, or more generally, for computing

---

[1] `https://github.com/NoahAmsel/PolarExpress`

the *polar decomposition* of a rectangular matrix. We apply our new `Polar Express` method (Algorithm 1, Implementation 1) to compute the descent direction in the increasingly popular `Muon` optimizer. In Figure 1, we show that using `Polar Express` within `Muon` consistently results in lower validation loss across all learning rates when training a GPT-2 model, as compared to other matrix sign methods (Cesista et al., 2025; Jordan et al., 2024b).

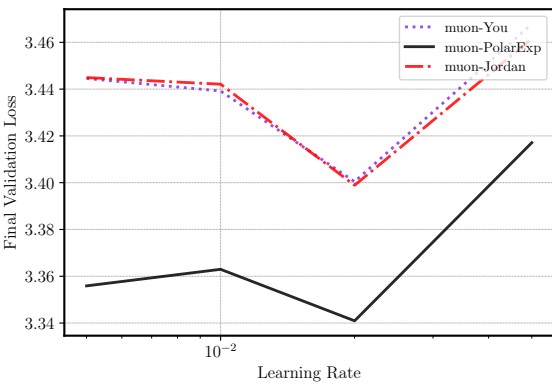 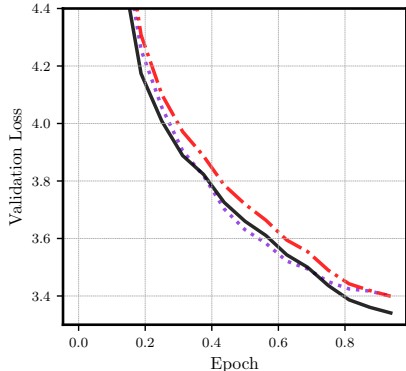

Figure 1: Training a GPT-2-Large model (774M params) on 1 billion tokens from the FineWeb dataset (Penedo et al., 2024). The label muon-`<name>` refers to implementing `Muon` using `<name>` to compute the polar factor. Left: final validation loss across learning rates. Right: validation loss across epochs using the best learning rate. The best learning rate ($lr$) and final validation loss for each method were `muon-You` ($lr = 0.02$): 3.399, `muon-Jordan` ($lr = 0.02$): 3.398 and `muon-PolarExp` ($lr = 0.02$): 3.340.

## 1.1 THE MUON METHOD

The `Muon` optimizer has recently gained popularity for training large language models, often outperforming state-of-the-art adaptive gradient methods like Adam and AdamW (Kingma & Ba, 2015; Loshchilov & Hutter, 2019). `Muon` has been used to set records for the NanoGPT speedrun (Jordan et al., 2024b), to expand the Pareto frontier of performance versus training FLOPs for large language models (Liu et al., 2025; Shah et al., 2025), and even to train a 1 trillion parameter frontier LLM (Kimi Team et al., 2025).

The `Muon` update rule (Bernstein & Newhouse, 2024b) is defined as follows. Let $\lambda, \beta > 0$ be the learning rate and momentum coefficient hyperparameters. (By default, $\beta = 0.9$.) Let $W_t \in \mathbb{R}^{m \times n}$ be the weight matrix of a given neural network layer at iteration $t$, and let $G_t \in \mathbb{R}^{m \times n}$ be its (stochastic) gradient. Let $M_t \in \mathbb{R}^{m \times n}$ be the running momentum estimate of the gradient, where $M_0 = 0$. The `Muon` update is given by

$$M_t = \beta M_{t-1} + (1 - \beta) G_t, \qquad W_{t+1} = W_t - \lambda \operatorname{polar}(M_t).$$

Whereas standard stochastic gradient descent (SGD) with momentum updates the weight matrix by taking a step in the direction $-M_t$, the `Muon` method steps in the direction $-\operatorname{polar}(M_t)$, where $\operatorname{polar}(M)$ denotes the closest semi-orthogonal matrix to $M$ (Higham, 2008, Chapter 8). Concretely, if $M = U \Sigma V^\mathsf{T}$ is the singular value decomposition (SVD) of $M$, then

$$\operatorname{polar}(M) := U V^\mathsf{T}. \tag{1}$$

The matrix $\operatorname{polar}(M)$ can be seen as a generalization of the matrix sign function to rectangular matrices (Benzi & Huang, 2019). Indeed, when $M$ is square symmetric with eigendecomposition $M = V \Lambda V^\mathsf{T}$, $\operatorname{polar}(M)$ exactly coincides with the matrix sign function $\operatorname{sign}(M) = V \operatorname{sign}(\Lambda) V^\mathsf{T}$ (Higham, 2008, Chapter 5). Equivalently, $\operatorname{polar}(M)$ is the left orthogonal factor of the polar decomposition of $M$ (Higham, 2008, Chapter 8). The motivation for `Muon` is that $-\operatorname{polar}(M)$ gives the steepest-descent direction with respect to the *spectral norm* (instead of the Frobenius norm, as in standard SGD). For analysis and further discussion on `Muon` we refer the reader to (Jordan et al., 2024b; Bernstein & Newhouse, 2024b; Pethick et al., 2025; Riabinin et al., 2025; Carlson et al., 2015a;b). In this paper, we take the `Muon` update rule as given and focus on the problem of efficiently computing the polar decomposition $\operatorname{polar}(M)$.

## 1.2 Computing the Polar Factor

Although $\mathrm{polar}(\boldsymbol{M})$ can be computed directly via an SVD in $O(mn\min(m,n))$ time, doing so is prohibitively expensive in deep learning applications, especially as standard SVD algorithms fail to take full advantage of the parallelism available on GPUs. There has been significant work on highly-parallel methods for the SVD, but the most common approaches actually require computing the matrix-sign function as a subroutine (Nakatsukasa & Freund, 2016; Nakatsukasa & Higham, 2013). Numerical analysts have spent decades developing iterative methods for computing $\mathrm{polar}(\boldsymbol{M})$. This rich line of work includes Newton-Schulz (Higham, 2008, Chapter 8), Padé iteration (Kenney & Laub, 1991; Higham, 1986), the Newton and scaled Newton iterations (Higham, 2008, Chapter 8), the QDWH iteration (Nakatsukasa et al., 2010; Nakatsukasa & Higham, 2013), and *Zolo-pd* (Nakatsukasa & Freund, 2016). Unfortunately, as discussed in Appendix B, most of these methods are based on rational approximations to the function $\mathrm{sign}(x)$ and require computing matrix inverses or QR decompositions. Such methods are ill-suited to GPU acceleration and deep learning applications. In contrast, the older Newton-Schulz method is based on *polynomial* approximation of $\mathrm{sign}(x)$ and uses only matrix-matrix products. Thus, `Muon` initially used Newton-Schulz (Bernstein & Newhouse, 2024a). Indeed, `Muon` stands for "MomentUm Orthogonalized by Newton-Schulz" (Jordan et al., 2024b). For a more comprehensive discussion on prior work, see Appendix B.

**The Newton-Schulz methods.** Newton-Schulz constructs a sequence of approximations $\boldsymbol{X}_t \approx \mathrm{polar}(\boldsymbol{M})$ as follows:

$$\boldsymbol{X}_0 = \boldsymbol{M}/\|\boldsymbol{M}\|_{\mathrm{F}}, \qquad \boldsymbol{X}_{t+1} = \frac{3}{2}\boldsymbol{X}_t - \frac{1}{2}\boldsymbol{X}_t\boldsymbol{X}_t^\top\boldsymbol{X}_t. \tag{2}$$

At each iteration, this rule effectively applies the cubic polynomial $p(x) = \frac{3}{2}x - \frac{1}{2}x^3$ to each singular value of $\boldsymbol{X}_t$. The scalar fixed-point iteration $x_{t+1} = p(x_t)$ converges to $\mathrm{sign}(x_0)$ as $t \to \infty$, provided $|x_0| \leq 1$. As a result, the matrix iteration satisfies $\lim_{t\to\infty} \boldsymbol{X}_t = \boldsymbol{U}\boldsymbol{V}^\top = \mathrm{polar}(\boldsymbol{X}_0)$. Higher-degree versions of Newton-Schulz follow the same principle. For example, the degree-5 polynomial $p(x) = (15x - 10x^3 + 3x^5)/8$ converges even faster. The Newton-Schulz iterations converge super-exponentially when $\boldsymbol{X}_t$ is sufficiently close to $\mathrm{polar}(\boldsymbol{M})$, but they suffer from slow initial convergence; when $\boldsymbol{X}_0$ is far from $\mathrm{polar}(\boldsymbol{M})$, the approximation improves slowly over the first few iterations. Due to the slow initial convergence of Newton-Schulz, Chen & Chow (2014) developed a version of the Newton-Schulz iteration, which adapts the polynomial at each iteration. The resulting method achieves a faster initial convergence, while retaining super-exponential convergence in later iterations. `Polar Express` is inspired by their method.

**The Jordan and You methods.** In `Muon`, high accuracy approximations to $\mathrm{polar}(\boldsymbol{M})$ are usually not necessary. The primary goal is instead to compute a coarse approximation in as few iterations as possible. To accelerate convergence in the low-accuracy regime, Jordan recently proposed a fixed-point iteration based on the polynomial $p(x) = 3.4445x - 4.7750x^3 + 2.0315x^5$, which was found using a heuristic numerical search (Jordan et al., 2024b). Unlike Newton-Schulz, the scheme that Jordan proposed does not converge to $\mathrm{polar}(\boldsymbol{M})$, but plateaus at an error of $\approx 0.3$. However, it reaches this level of accuracy rapidly and outperforms the Newton-Schulz when only a small number of iterations are performed. Building on this idea, You proposed a method that applies six different polynomial updates in succession, which were again found by heuristic search. This method achieves better accuracy than Jordan's but still fails to converge (Cesista et al., 2025).

## 1.3 Contributions

We present `Polar Express` (Algorithm 1), an iterative method for approximating $\mathrm{polar}(\boldsymbol{M})$. Our method dynamically adapts the polynomial update rule at each iteration, prioritizing rapid progress in the initial stage and high accuracy in the later stage. `Polar Express` constructs polynomials $p_1, \ldots, p_T$ so that the resulting composition is the optimal approximation to the sign function with respect to the supremum ($L^\infty$) norm (Theorem 3.1). By iteratively applying these polynomials to $\boldsymbol{M}$, `Polar Express` computes an approximation to $\mathrm{polar}(\boldsymbol{M})$ that is optimal in the worst-case. Our method converges to $\mathrm{polar}(\boldsymbol{M})$ super-exponentially (Theorem 3.3), and it quickly reaches a good approximation within just five to ten iterations. This early-stage acceleration is especially valuable in deep learning applications, where runtime efficiency takes precedence over

high accuracy. In contrast, classical methods like Newton-Schulz suffer from a slow initial convergence, while recent heuristic proposals (Jordan et al., 2024b; Cesista et al., 2025) fail to converge. Our method is efficient to run on GPUs, using only a few matrix-matrix products per iteration. We give an explicit instantiation of `Polar Express` in Implementation 1, which incorporates minor modifications to make it compatible with half-precision arithmetic (see Section 3.4). Implementation 1 is very short and easy to use, with no dependencies except PyTorch. It serves as a drop-in replacement for previous methods. In numerical experiments, `Polar Express` outperforms previous methods on synthetic matrices and gradient matrices from a GPT-2 transformer (Figure 3). We demonstrate the effectiveness of using `Polar Express` within the `Muon` optimizer in Figure 1, showing that it consistently improves the training of GPT-2 language models on 1 billion tokens of the FineWeb dataset (Penedo et al., 2024). Our method has been adopted into the NanoGPT speedrun (Jordan et al., 2024a), a heavily optimized implementation that serves as a benchmark for LLM training efficiency.

**Notation.** We let $\|\boldsymbol{M}\|_{\mathrm{F}}$ and $\|\boldsymbol{M}\|_2$ denote the Frobenius norm and spectral norm (largest singular value) of a matrix $\boldsymbol{M}$, respectively. We denote the spectrum (set of singular values) by $\sigma(\boldsymbol{M})$. Let $\mathbb{P}_d$ be the set of polynomials of degree at most $d$. For odd $d$, $\mathbb{P}_d^{\mathrm{odd}}$ denotes the set of polynomials of degree at most $d$ containing only odd-degree monomials. For a polynomial $p$, $\deg(p)$ is its degree. Let $\mathrm{sign}(x)$ be the scalar sign function, which satisfies $\mathrm{sign}(0) = 0$, $\mathrm{sign}(x) = 1$ if $x > 0$ and $\mathrm{sign}(x) = -1$ if $x < 0$. For a polynomial $p \in \mathbb{P}_d^{\mathrm{odd}}$ and a matrix $\boldsymbol{M}$ with rank reduced SVD given by $\boldsymbol{M} = \boldsymbol{U}\boldsymbol{\Sigma}\boldsymbol{V}^{\mathsf{T}}$ and positive singular values $\sigma_1 \geq \cdots \geq \sigma_{\mathrm{rank}(\boldsymbol{M})} > 0$, we define $p(\boldsymbol{M}) := \boldsymbol{U}p(\boldsymbol{\Sigma})\boldsymbol{V}^{\mathsf{T}}$, where $p(\boldsymbol{\Sigma})$ is the diagonal matrix with diagonal entries $p(\sigma_i)$ for $i = 1, \ldots, \mathrm{rank}(\boldsymbol{M})$.

## 2 APPROXIMATIONS BY COMPOSITIONS OF POLYNOMIALS

To design a GPU-friendly method for computing $\mathrm{polar}(\boldsymbol{M})$, we limit ourselves to the following GPU-friendly operations: (i) linear combinations of matrices (given scalars $\beta, \gamma \in \mathbb{R}$ and matrices $\boldsymbol{B}$ and $\boldsymbol{C}$, compute $\beta\boldsymbol{B} + \gamma\boldsymbol{C}$) and (ii) matrix-matrix products (compute $\boldsymbol{B}\boldsymbol{C}$). While both these computational primitives are well-suited for parallel computing environments, matrix-matrix products come at a higher computational cost than linear combinations. Therefore, our method attempts to minimize the number of matrix-matrix products. A key observation is that we can compute *odd* monomials of $\boldsymbol{M} = \boldsymbol{U}\boldsymbol{\Sigma}\boldsymbol{V}^{\mathsf{T}}$ using the following formula: $\boldsymbol{M}^{2q+1} := \boldsymbol{U}\boldsymbol{\Sigma}^{2q+1}\boldsymbol{V}^{\mathsf{T}} = \boldsymbol{M}(\boldsymbol{M}^{\mathsf{T}}\boldsymbol{M})^q$.[2] Hence, for an odd polynomial $p(x) = a_0 x + a_1 x^3 + \cdots + a_q x^{2q+1}$ we can compute

$$p(\boldsymbol{M}) := a_0\boldsymbol{M} + a_1\boldsymbol{M}(\boldsymbol{M}^{\mathsf{T}}\boldsymbol{M}) + \cdots + a_q\boldsymbol{M}(\boldsymbol{M}^{\mathsf{T}}\boldsymbol{M})^q.$$

It has been shown that for an arbitrary polynomial $p$, one requires $\Theta(\deg(p)^{1/2})$ products to compute $p(\boldsymbol{M})$ (Paterson & Stockmeyer, 1973); see also Jarlebring & Lorentzon (2025) for related work. This compares favorably to the naive approach that forms all monomials in $p$ and then sums them together, which requires $\Omega(\deg(p))$ products. However, if $p$ can be expressed as a composition of $T$ polynomials, each of degree $d$

$$p = p_T \circ p_{T-1} \circ \cdots \circ p_1, \tag{3}$$

then the degree of $p$ is $d^T$, and $p(\boldsymbol{M})$ can be efficiently computed recursively by

$$\boldsymbol{X}_0 = \boldsymbol{M}, \quad \boldsymbol{X}_t = p_t(\boldsymbol{X}_{t-1}) \text{ for } t = 1, 2, \ldots, T. \tag{4}$$

The final iterate is $\boldsymbol{X}_T = p(\boldsymbol{M})$, which we compute with just $O(Td)$ matrix-matrix products. Iterative methods for $\mathrm{polar}(\boldsymbol{M})$ can be seen in this light. For instance, the degree-5 Newton-Schulz method uses the polynomial update $p_t(x) = \frac{15}{8}x - \frac{10}{8}x^3 + \frac{3}{8}x^5$ for each $t = 1, \ldots, T$. The composition $p = p_T \circ \cdots \circ p_1$ approximates $\mathrm{sign}(x)$, and the approximation error goes to 0 as $T$ grows. In this paper, we ask the following question: what choice of $p_T \circ \cdots \circ p_1$ gives the *best* approximation to $\mathrm{sign}(x)$?

The method we will present is optimal in the following sense: given lower and upper bounds $\ell$ and $u$ on the singular values of $\boldsymbol{M}$, an odd degree $d \in \mathbb{N}$, and the number of iterations $T \in \mathbb{N}$, our method computes the composition $p^\star(\boldsymbol{M})$ that minimizes the worst-case error in the spectral norm. That is,

$$p^\star = \underset{\substack{p = p_T \circ p_{T-1} \circ \cdots \circ p_1 \\ p_t \in \mathbb{P}_d^{\mathrm{odd}}}}{\arg\min} \; \underset{\substack{\boldsymbol{M} \in \mathbb{R}^{m \times n} \\ \sigma(\boldsymbol{M}) \subset [\ell, u]}}{\max} \; \|\mathrm{polar}(\boldsymbol{M}) - p(\boldsymbol{M})\|_2. \tag{5}$$

---

[2]For non-symmetric matrices, e.g. rectangular matrices, we cannot compute even polynomials of the singular values without first explicitly computing the SVD. We are therefore restricted to odd polynomials.

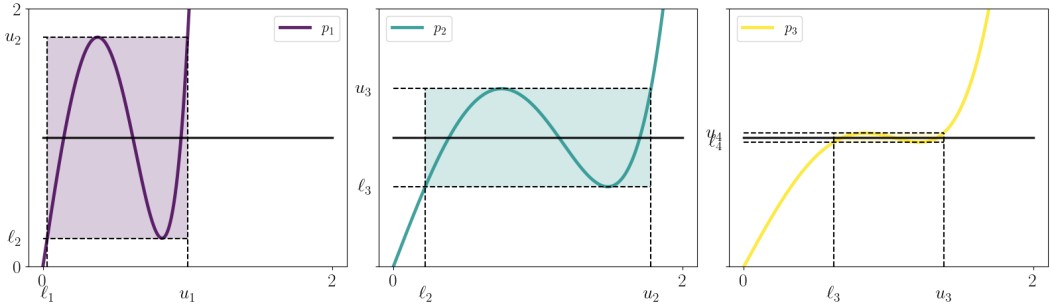

Figure 2: The evolution of the first three optimal polynomials $p_1$, $p_2$, and $p_3$ and the corresponding lower bounds $\ell_{t+1} = p_t(\ell_t)$ and upper bounds $u_{t+1} = 2 - \ell_{t+1}$, as described in Theorem 3.1. The horizontal black line shows $y = 1$. The polynomial degree is $d = 5$. We set $\ell_1 = 0.03$ and $u_1 = 1$.

Given that $\mathrm{polar}(\boldsymbol{M}) - p(\boldsymbol{M}) = \boldsymbol{U}(\boldsymbol{I} - p(\boldsymbol{\Sigma}))\boldsymbol{V}^\top$, and by the unitary invariance of the spectral norm, we have that (5) is equivalent to[3]

$$p^\star = \operatorname*{arg\,min}_{\substack{p = p_T \circ p_{T-1} \circ \cdots \circ p_1 \\ p_t \in \mathbb{P}_d^{\mathrm{odd}}}} \max_{x \in [\ell, u]} |1 - p(x)|. \tag{6}$$

In other words, the problem given in (5) reduces to that of finding a "uniform" approximation to the constant function $x \mapsto 1$ over the interval $[\ell, u]$, as given in (6). Uniform approximation on an interval by polynomials or rational functions of a given degree is a central topic in approximation theory (Trefethen, 2020). Here, we seek an approximation of a particular form—a *composition* of odd polynomials of fixed degrees. In the next section, we solve the optimization problem of (6) and use the solution to create `Polar Express`.

## 3   THE POLAR EXPRESS

### 3.1   GREEDY IS OPTIMAL

The key observation is that the polynomial used in each iteration can be chosen greedily, given the choice of polynomials from the previous iterations. For the first iteration, we choose $p_1$ so as to map the interval $[\ell, u]$ as close to 1 as possible. That is, it minimizes $\max_{x \in [\ell, u]} |1 - p_1(x)|$. The image of $p_1$ will be a new interval $[\ell_2, u_2]$, where

$$\ell_2 = \min_{x \in [\ell, u]} p_1(x) \qquad u_2 = \max_{x \in [\ell, u]} p_1(x) \tag{7}$$

We now pick $p_2$ to map the interval $[\ell_2, u_2]$ as close to 1 as possible, obtaining a new interval $[\ell_3, u_3]$ that is the image of $[\ell, u]$ through $p_2 \circ p_1$. We continue this process for as many iterations as desired.

The following theorem guarantees that this process finds the solution to (6), and thereby also (5). The scheme is also outlined in Figure 2, which demonstrates the evolution of the lower bounds $\ell_t$, the upper bounds $u_t$, and the polynomials $p_t$ across iterations. The proof is in Appendix C.

**Theorem 3.1.** Let $d$ be odd and define $\ell_1 = \ell$ and $u_1 = u$. For $t = 1, \ldots, T$ define

$$p_t = \operatorname*{arg\,min}_{p \in \mathbb{P}_d^{\mathrm{odd}}} \max_{x \in [\ell_t, u_t]} |1 - p(x)|, \quad \ell_{t+1} = \min_{x \in [\ell_t, u_t]} p_t(x), \quad u_{t+1} = \max_{x \in [\ell_t, u_t]} p_t(x) \tag{8}$$

The resulting composition $p^\star := p_T \circ p_{T-1} \circ \cdots \circ p_1$ is optimal and the error is given by:

$$\max_{x \in [\ell, u]} |1 - p^\star(x)| = \operatorname*{min}_{\substack{p = p_T \circ p_{T-1} \circ \cdots \circ p_1 \\ p_t \in \mathbb{P}_d^{\mathrm{odd}}}} \max_{x \in [\ell, u]} |1 - p(x)| = 1 - \ell_{T+1}. \tag{9}$$

---

[3]For completeness, the equivalence between (5) and (6) is proven in Appendix E.

Furthermore the new error, lower and upper bounds can be computed through

$$\ell_{t+1} = p_t(\ell_t), \quad u_{t+1} = 2 - \ell_{t+1}, \quad \text{and} \quad \max_{x \in [\ell_t, u_t]} |1 - p_t(x)| = 1 - \ell_{t+1}. \tag{10}$$

**Remark 3.2** (Why a fixed degree?). We note that choice of the degree of each $p_1, p_2, \ldots, p_T$ need not be the same for Theorem 3.1 to hold. More generally, one may specify a sequence of degrees $d_1, \ldots, d_T$ and define each $p_t$ as $p_t = \arg\min_{p \in \mathbb{P}_{d_t}^{\text{odd}}} \max_{x \in [\ell_t, u_t]} |p(x) - 1|$ for $t = 1, \ldots, T$. However, Lee et al. (2022, Table 2) supports setting $d_t = 5$, as we do.

Fortunately, (10) shows that once $p_t$ has been found, we can compute the new lower and upper bounds $\ell_{t+1}$ and $u_{t+1}$ simply by evaluating $p_t(\ell_t)$. Hence, for any *fixed* upper and lower bounds on the singular values of $M$, we can *precompute* all the polynomials $p_1, \ldots, p_T$ and the bounds $[\ell_1, u_1], \ldots, [\ell_{T+1}, u_{T+1}]$. Then, applying the iterative procedure of (4), the final iterate $X_T$ will satisfy the following error bound:

$$\| \text{polar}(M) - X_T \|_2 = \| \text{polar}(M) - p^\star(M) \|_2 \leq 1 - \ell_{T+1}. \tag{11}$$

From the optimality guarantee of Theorem 3.1, we know that our method converges at least as fast as the Newton-Schulz iteration of the same degree. Combining this fact with an existing analysis of Newton-Schulz, we immediately get the following convergence guarantee showing that our method enjoys faster than exponential convergence. The proof can be found in Appendix D.

**Theorem 3.3.** Let $M$ be a matrix normalized so that $\sigma(M) \subset [\ell, 1]$. Let $X_T = p^\star(M)$, where $p^\star$ is the polynomial from Theorem 3.1 with $d = 2q + 1$. Then, we have

$$\| \text{polar}(M) - X_T \|_2 \leq |1 - \ell^2|^{(q+1)^T}. \tag{12}$$

Hence, for $d = 3$ and $d = 5$ the method converges quadratically and cubically, respectively.

In fact, our method is strictly faster than Newton-Schulz, even if $\sigma_{\min}(M) < \ell$. When $\sigma_{\min} = \ell$, `Polar Express` is about twice as fast as Newton-Schulz (cf. Chen & Chow (2014, Section 3.1)). Recent work has analyzed the stability and convergence of `Muon` when the polar factor is computed inexactly (Shulgin et al., 2025; Refael et al., 2025). Combining these analyses with Theorem 3.3 immediately yields a convergence guarantee for `Muon` as implemented with `Polar Express`.

## 3.2 FINDING THE OPTIMAL POLYNOMIAL FOR EACH ITERATION

Theorem 3.1 shows that we can solve (6) by greedily choosing the optimal approximation $p_t \in \mathbb{P}_d^{\text{odd}}$ for each interval $[\ell_t, u_t]$ for $t = 1, \ldots, T$. In this section, we show how to find each $p_t$. Since we are now focused on just one iteration, we drop the subscripts. Given $\ell$ and $u$, we wish to solve the following optimization problem:

$$\arg\min_{p \in \mathbb{P}_d^{\text{odd}}} \max_{x \in [\ell, u]} |1 - p(x)| \tag{13}$$

That is, we seek a minimax or uniform approximation of the function $x \mapsto 1$ on $[\ell, u]$ from the set of odd polynomials. (Equivalently, we seek a minimax optimal approximation to $\text{sign}(x)$ on $[-u, -\ell] \cup [\ell, u]$.) Problems of this form are well-studied in approximation theory and numerical analysis. The key mathematical insight underlying their solution is the Equioscillation Theorem, which we state formally for our setting in Lemma C.1. This theorem is the basis of the Remez algorithm (Pachón & Trefethen, 2009; Parks & McClellan, 1972), a general-purpose method that finds a (nearly) optimal polynomial approximation of a given degree to *any* function on any interval. With a very minor modification to handle the constraint that $p$ be odd, Remez can solve (13).

However, the Remez algorithm is complicated and notoriously difficult to implement correctly.[4] Fortunately, we do not need the algorithm in its full generality; we seek only low-degree polynomial approximations, and the function we wish to approximate is just $f(x) = 1$. We use the Equioscillation Theorem to derive (17), an explicit, closed-form solution to (13) for the degree $d = 3$ case. Up

---

[4] For implementations of the general Remez algorithm, we recommend Chebfun or `lolremez`.

to rescaling, this turns out to be the same polynomial derived by different means in Chen & Chow (2014). For $d = 5$, we present Algorithm 2, a simpler way of solving (13) that is mathematically equivalent to Remez in our setting. This algorithm is implemented in its entirety in Implementation 2. For more details, we refer the reader to Appendix F.

## 3.3 UPPER AND LOWER BOUNDS ON THE SINGULAR VALUES

To instantiate our method, we need upper and lower bounds $u$ and $\ell$ on the singular values of the input matrix $M$. A trivial upper bound is given by $\|M\|_{\mathrm{F}}$. This can be quite loose in the worst case. In practice, it is off only by a small constant factor because the gradient matrices of the weights of dense linear layers in neural networks tend to have small effective rank (Yang et al., 2024). We therefore rescale $M$ by $\|M\|_{\mathrm{F}}$ and set $u = 1$. It is difficult to efficiently find a good lower bound on $\sigma_{\min}$, so we are forced to guess. Fortunately, the consequences of a bad guess are not severe. The method converges for any $\ell \in (0, u]$, and even an order of magnitude error only delays convergence by a few iterations. For matrices stored in floating point arithmetic, the singular values are usually larger than machine precision $\epsilon_{\mathrm{mach}}$ (Boutsikas et al., 2024). We work in bfloat16, which has $\epsilon_{\mathrm{mach}} = 2^{-8} \approx 3.91 \cdot 10^{-3}$, so we set $\ell = 10^{-3}$. Since we use these bounds for all input matrices, we can pre-compute the optimal polynomials once and apply them to as many inputs as we want.

## 3.4 FINITE PRECISION CONSIDERATIONS

When working in finite-precision arithmetic, especially the half-precision bfloat16 format used in deep learning, we must take some care to avoid blowups and other problems due to numerical error. To this end, we make a few small but crucial changes to the method in the offline stage that stabilize it with a negligible effect on accuracy. One issue arises when numerical round-off creates singular values that are slightly larger than our current upper bound $u_t$. To fix it, we replace each polynomial $p_t$ by $x \mapsto p_t(x/1.01)$, effectively increasing $u_t$. Another issue, identified by Nakatsukasa & Higham (2013), is due to the non-monotonicity of $p_t$. We address it by using slightly suboptimal (but less oscillatory) polynomials in the early iterations, as suggested by Chen & Chow (2014). For a detailed discussion on the finite precision considerations, we refer to Appendix G.

## 3.5 THE ALGORITHM

---

**Algorithm 1** The General `Polar Express`

**input:** Matrix $M$, iteration count $T$, degree $d$, approximate lower bound $\ell$.
**output:** An approximation $X_T$ to polar($M$).

1  | Offline: precompute polynomials in `float64`
2  | $\ell_1 = \ell$, $u_1 = 1$.
3  | **for** $t = 1, 2, \ldots, T$ **do**
4  |     Solve using Remez (Appendix F):
   |     $p_t = \underset{p \in \mathbb{P}_d^{\mathrm{odd}}}{\arg\min} \; \underset{x \in [\max(\ell_t, u_t/10), \, u_t]}{\max} |1 - p(x)|$
5  |     $p_t \leftarrow p_t(\cdot/1.01)$
6  |     $\ell_{t+1} \leftarrow p_t(\ell_t), \quad u_{t+1} \leftarrow 2 - \ell_{t+1}$
7  | **end for**
8  |
9  | Online: apply precomputed polynomials in `bfloat16`
10 | Set $X_0 = M/(\|M\|_{\mathrm{F}} + 10^{-2})$.
11 | **for** $t = 1, 2, \ldots, T$ **do**
12 |     $X_t = p_t(X_{t-1})$
13 | **end for**
14 | **return** $X_T$.

---

We give the pseudocode of our proposed method for any degree in Algorithm 1. We give the specific Python code of the `Polar Express` with degree $d = 5$ and $\ell = 10^{-3}$ used in our GPT experiments in Implementations 1 and 2 in Appendix A. Both incorporate the finite precision considerations discussed in Section 3.4. Our algorithm precomputes the polynomials $p_1, \ldots, p_T$ of Theorem 3.1 in full precision using the results of Section 3.2 (or the Remez algorithm for $d > 5$). This stage is offline because the coefficients of the polynomials are only computed and stored once. For every subsequent call to the algorithm, these coefficients are reused and the offline stage is skipped. For instance, in Implementation 1 these polynomials have been precomputed and stored in the variable `coeffs_list`.

The online stage can be performed in lower precision (bfloat16) for greater speed on a GPU. Horner's rule can be used to carry out each iteration. For instance, if $p_t = ax + bx^3 + cx^5$, then

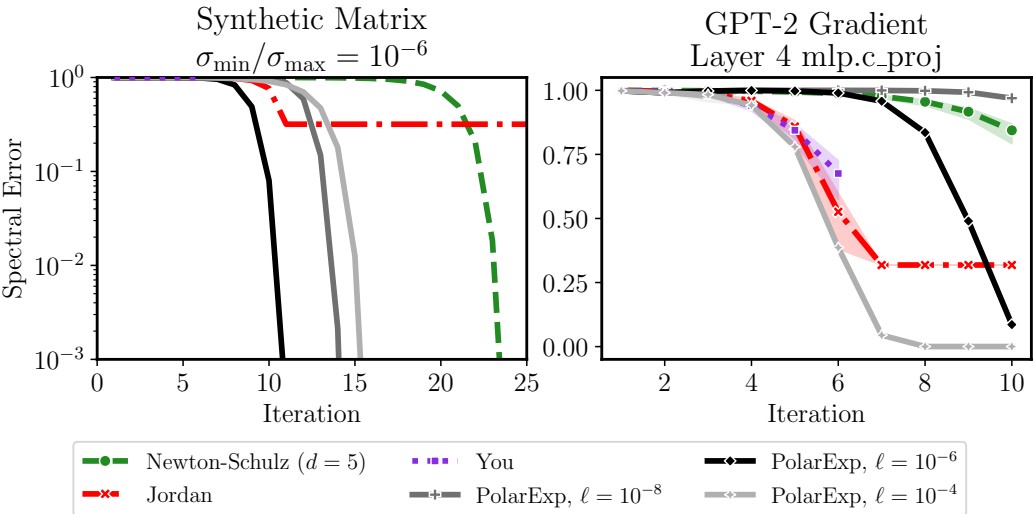

Figure 3: Convergence of degree-5 polynomial methods. Polar Express outperforms other methods at every iteration when tuned properly. Left panel: synthetic matrix with $\sigma_{\max} = 1$, $\sigma_{\min} = 10^{-6}$. Right panel: gradient from randomly-initialized GPT-2 model on a batch of language modeling data. Shaded region shows 90% interval over 20 batches of data.

$X_t = X_{t-1}(aI + Y_{t-1}(bI + cY_{t-1}))$ where $Y_{t-1} = X_{t-1}^\top X_{t-1}$. A simple implementation of the offline stage of Algorithm 1 is given in Implementation 2. For deep learning applications, we recommend using $d = 5$ and $T = 5$ or 6 with $\ell_1 = 10^{-3}$. With these parameters, the offline stage as implemented in Implementation 2 gives the polynomials encoded in `coeffs_list` in Implementation 1. All told, our proposal for `Muon` is to apply the composition of these polynomials to $M/(\|M\|_F + 10^{-2})$.[5]

## 4 NUMERICAL EXPERIMENTS

### 4.1 CONVERGENCE OF POLAR EXPRESS

We compare `Polar Express` against degree-5 Newton-Schulz and the methods of Jordan et al. (2024b) and Cesista et al. (2025). We first generate a random matrix whose singular values are evenly spaced on a logarithmic scale between $10^{-6}$ and 1, with singular vectors chosen randomly. The left panel of Figure 3 shows the results. Since all the methods in this plot use degree-5 polynomials, their computational and runtime costs are all proportional to the number of iterations. As expected, Newton-Schulz converges but makes almost no progress for the first 17 iterations. Jordan's method rapidly achieves an error of $\approx 0.3$ after just 11 iterations, but ceases to converge further. You's method, which is only defined for six iterations, converges at a similar rate as Jordan's method. When `Polar Express` is instantiated with $\ell = \sigma_{\min}$, it dominates the other methods at every iteration, achieving excellent accuracy after just 11 iterations and converging about twice as fast as Newton-Schulz to any given error. Even when $\ell$ is wrong by two orders of magnitude in either direction, the method remains competitive, though it does not outperform Jordan's method until iteration 13 or 14. We also test convergence on a non-synthetic matrix: the gradient of a weight matrix from the fourth transformer block of a GPT-2 model (Figure 3, right). Again, the best-tuned version of `Polar Express` outperforms the other methods, but setting $\ell$ to be many orders of magnitude too small can delay convergence. Note that Figure 3 measures error in the spectral norm. For many applications we may be satisfied with a looser measure of error; see Appendix H.1.

---

[5]In Appendices I and J, we describe two further algorithmic ideas. They are not used in our `Muon` experiments but they may be beneficial in other settings, and we believe they merit further study.

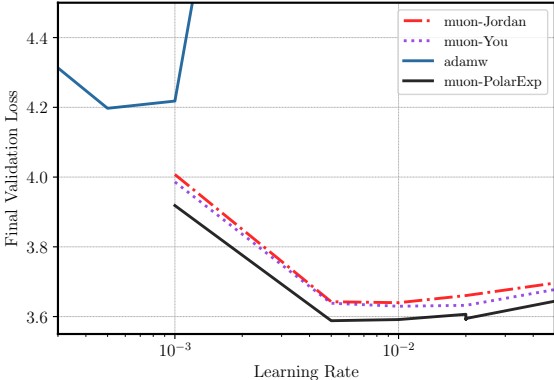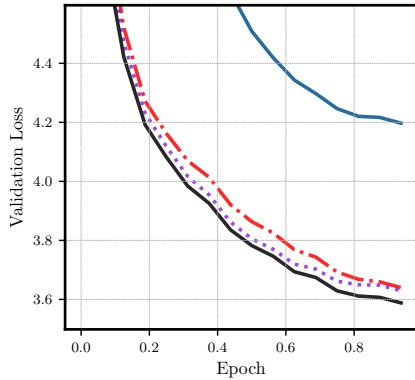

Figure 4: Training a GPT-2-Small (124M) model on 1 Billion tokens of the FineWeb data set (Penedo et al., 2024). muon-<method> denotes Muon with 5 iterations of <method> to compute $\text{polar}(M)$. No weight decay is used. Left: final validation loss vs. learning rate. The best final validation losses for each method were adamw(lr =0.0005): 4.197, muon-Jordan(lr =0.01): 3.639, muon-You(lr =0.01): 3.629 and muon-PolarExp(lr =0.005): 3.588. Right: Validation loss vs. training iteration.

## 4.2 TRAINING GPT-2

We compare the performance of using Polar Express (Implementation 1) inside Muon against Jordan's (Jordan et al., 2024b) and You's (Cesista et al., 2025) methods. We train two architectures: GPT-2-Small ($n_{\text{embd}} = 768, n_{\text{layer}} = 12, n_{\text{head}} = 12$) and GPT-2-Large ($n_{\text{embd}} = 1280, n_{\text{layer}} = 36, n_{\text{head}} = 20$), both with a vocabulary size of 50,257 and a context length of 1024. We train on 1B tokens of the FineWeb dataset (Penedo et al., 2024) for one epoch with batch size 32. All runs use mixed precision (bfloat16) on 4 H100 GPUs with the learning rate schedule proposed in Jordan et al. (2024a)—a constant phase for the first 40% of training steps followed by linear decay. All methods for the matrix sign computations are performed in bfloat16 precision and use five iterations. Following nano-gpt (Jordan et al., 2024a), we assign Muon to all parameters with at least two dimensions (e.g., excluding RMS norm parameters), except for embeddings, unembeddings, and positional encodings. These excluded parameters are optimized with AdamW.[6]

Figures 1 and 4 show the resulting in terms of validation loss for the GPT-Large and GPT-Small models, respectively. In both cases, muon-PolarExp achieves a better validation loss than muon-Jordan or muon-You. The advantage is remarkably consistent across all learning rates and epochs. While not shown in Figures 1 and 4, muon-PolarExp also achieves a better training loss than the baselines, and the improvements in training loss are nearly identical to the improvements in validation loss. Furthermore, since all three of these matrix sign methods are equally expensive (they all apply a degree 5 polynomial at each iteration), improved validation loss in terms of training steps also implies improved loss in terms of wall clock time. For figures displaying the improvements in training loss and wall-clock time, see Appendix H.2, Figure 11.

## 4.3 ABLATIONS

**Accuracy of polar approximation** We now explore how the accuracy of approximating $\text{polar}(M)$ affects the optimization quality of Muon. Our main experiments with GPT-2 use 5 iterations. We trained GPT-2 Small with Muon using between 2 and 30 iterations of Polar Express instead. For comparison, we also implemented Muon with the *exact* polar factor, computed using torch.linalg.svd. Figure 5 shows the results. The left plot shows that when using only 2 or 3 iterations of Polar Express, the final validation loss is worse than when using 5 or 6 iterations. However, increasing the accuracy of the polar approximation further—even computing it exactly

---

[6]Code for our LLM training experiments is available at https://github.com/modichirag/GPT-opt/tree/polar, in the polar branch.

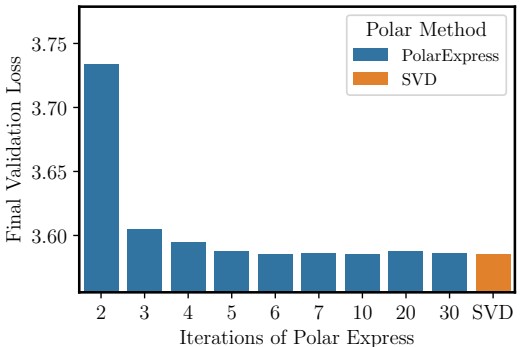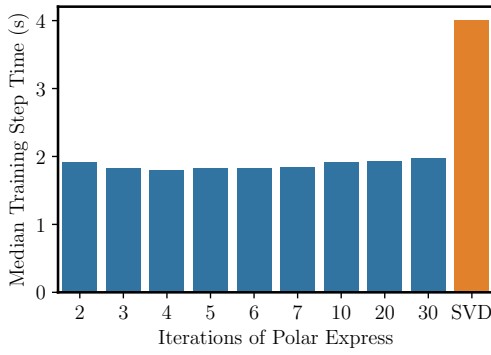

Figure 5: Ablating the number of iterations of `Polar Express` used to implement `Muon`, and comparing to computing $\mathrm{polar}(M)$ exactly via an SVD. Left: using $> 6$ iterations or the SVD does not improve final validation loss. Right: Runtime of `Muon` is not sensitive to the number of iterations of `Polar Express`, but the SVD makes it significantly slower. All runs use GPT-2-Small with 1 Billion tokens of FineWeb data, learning rate 0.05, and weight decay 0.1.

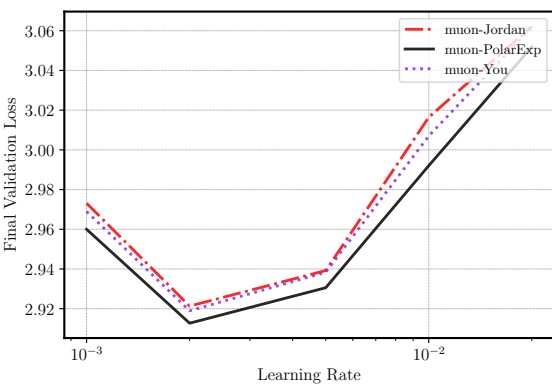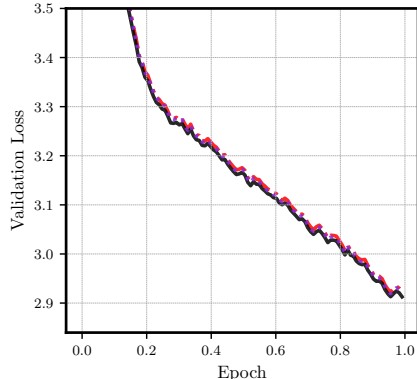

Figure 6: Training GPT-2-Large on 10 billion tokens of FineWeb with weight decay 0.1. Best final validation losses were `muon-Jordan` (lr = 0.002): 2.921, `muon-You` (lr = 0.002): 2.919 and `muon-PolarExp` (lr = 0.002): 2.913.

with the SVD—does not improve the optimization quality. The right plot shows that changing the number of iterations does not meaningfully change the runtime of `Muon`; in our setting, the runtime of computing $\mathrm{polar}(M)$ is dominated by the forward and backward passes. However, the SVD is so costly that using it *doubles* the runtime of each training step. These results validate the standard way of implementing `Muon`: using 5 or 6 iterations of an iterative approximation like `Polar Express` rather than computing $\mathrm{polar}(M)$ exactly. For further experiments supporting this conclusion, see Appendix H.1, Figure 9.

**Weight decay** We also experimented with adding weight decay of 0.1 to the GPT-2 training runs, keeping all else the same. The results are presented in Appendix H.2, Figure 12. They are quite similar to Figures 1 and 4. We again find that `muon-PolarExp` outperforms the other methods.

**Number of Training Tokens** Our main experiments with GPT-2 use 1 billion tokens of training data from FineWeb (Penedo et al., 2024). We now select a subset of our training runs and extend them to 10 billion tokens. 10 billion tokens roughly matches the Chinchilla scaling rule for GPT-2-Large (774M params) and exceeds it for GPT-2-Small, as per Table 3 in Hoffmann et al. (2022). Figure 6 shows the results for GPT-2-Large with weight decay. (For GPT-2-Small, see Appendix H.2, Figure 13b). `Polar Express` still outperforms the baselines by a small but consistent margin.

**Acknowledgments**    This work was partially supported by NSF awards 2045590 and 2234660.

**Reproducibility statement**    A complete Pytorch implementation of our method is given in Appendix A. Details of our experiments, including hyperparameters, are given in Sections 4.1 and 4.2. Source code to reproduce our experiments is given in the supplementary materials and is available at `https://github.com/modichirag/GPT-opt/tree/polar`, in the `polar` branch. Proofs of all theoretical claims can be found in the appendices.

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

CONTENTS

## A CODE FOR POLAR EXPRESS

Implementation 1 gives a Python implementation of the online stage of Algorithm 1 for degree = 5, which we use in our numerical experiments. It uses hard-coded polynomials generated from Implementation 2 and incorporates a numerical safety factor of 1.01 as described in Section 3.4. This implementation is designed for ease of use. It is short, it has no dependencies besides PyTorch, and it is a drop-in replacement for previous implementations of matrix sign methods (Cesista et al., 2025; Jordan et al., 2024b), such as Modula (2024).[7]

**Implementation 1** Python code for `Polar Express` of degree = 5.

```python
from itertools import repeat
import torch

coeffs_list = [
    (8.28721201814563, -23.59588651909837, 17.300387312530933),
    (4.107059111542203, -2.9478499167379106, 0.5448431082926601),
    (3.9486908534822946, -2.908902115962949, 0.5518191394370137),
    (3.3184196573706015, -2.488488024314874, 0.51004894012372),
    (2.300652019954817, -1.6689039845747493, 0.4188073119525673),
    (1.891301407787398, -1.2679958271945868, 0.37680408948524835),
    (1.8750014808534479, -1.2500016453999487, 0.3750001645474248),
    (1.875, -1.25, 0.375),  # subsequent coeffs equal this numerically
]
# safety factor for numerical stability (but exclude last polynomial)
coeffs_list = [(a / 1.01, b / 1.01**3, c / 1.01**5)
               for (a, b, c) in coeffs_list[:-1]] + [coeffs_list[-1]]

@torch.compile
def PolarExpress(G: torch.Tensor, steps: int) -> torch.Tensor:
    assert G.ndim >= 2
    X = G.bfloat16()  # for speed
    if G.size(-2) > G.size(-1): X = X.mT  # this reduces FLOPs
    X = X / (X.norm(dim=(-2, -1), keepdim=True) * 1.01 +1e-7)
    hs = coeffs_list[:steps] + list(
        repeat(coeffs_list[-1], steps - len(coeffs_list)))
    for a, b, c in hs:
        A = X @ X.mT
        B = b * A + c * A @ A
        X = a * X + B @ X  # X <- aX + bX^3 + cX^5
    if G.size(-2) > G.size(-1): X = X.mT
    return X
```

Implementation 2 gives a Python implementation of the offline stage of Algorithm 1. This code was used to construct the coefficients of the polynomials given in Implementation 1, which in turn were used in our `Muon` experiments (Section 4.2). It uses $\ell = 10^{-3}$ and $u = 1$ by default. It incorporates Algorithm 2 and the finite precision modifications described in Section 3.4.

**Implementation 2** `Polar Express`, Offline Stage

```python
from math import inf, sqrt
import numpy as np

def optimal_quintic(l, u):
    assert 0 <= l <= u
    if 1 - 5e-6 <= l / u:
        # Above this threshold, the equioscillating polynomials
        # is numerically equal to...
        return (15/8)/u, (-10/8)/(u**3), (3/8)/(u**5)
    # This initialization becomes exact as l -> u
    q = (3*l + u) / 4
```

---

[7]Code including Implementations 1 and 2 can also be found at `https://github.com/NoahAmsel/PolarExpress`.

```
    r = (l + 3*u) / 4
    E, old_E = inf, None
    while not old_E or abs(old_E - E) > 1e-15:
        old_E = E
        LHS = np.array([
            [l, l**3, l**5, 1],
            [q, q**3, q**5, -1],
            [r, r**3, r**5, 1],
            [u, u**3, u**5, -1],
        ])
        a, b, c, E = np.linalg.solve(LHS, np.ones(4))
        q, r = np.sqrt((-3*b + np.array([-1, 1]) *
                        sqrt(9*b**2 - 20*a*c)) / (10*c))
    return float(a), float(b), float(c)

def optimal_composition(l, num_iters, cushion=0.02407327424182761):
    u = 1
    coefficients = []
    for _ in range(num_iters):
        a, b, c = optimal_quintic(max(l, cushion*u), u)
        # Due to cushioning, this may be centered around 1 with
        # respect to 0.024*u, u. Recenter it around 1 with respect
        # to l, u, meaning find c so that 1 - c*p(l) = c*p(u) - 1:
        pl = a*l + b*l**3 + c*l**5
        pu = a*u + b*u**3 + c*u**5
        rescalar = 2/(pl + pu)
        a *= rescalar; b *= rescalar; c *= rescalar
        # Optionally incorporate safety factor here:
        # a /= 1.01; b /= 1.01**3; c /= 1.01**5
        coefficients.append((a, b, c))
        l = a*l + b*l**3 + c*l**5
        u = 2 - l
    return coefficients

print(*optimal_composition(1e-3, 10), sep="\n")
```

# B  RELATED WORK

Computing $\mathrm{polar}(M)$ is an important and longstanding problem in numerical linear algebra, with applications spanning electronic structure calculations, lattice quantum chromodynamics, orthogonal Procrustes analysis, parallel algorithms for computing the SVD, and beyond; see e.g. (Higham, 1986; Kaneko et al., 2013; Douglas Carroll & Arabie, 1998; Gower & Dijksterhuis, 2004; Neuberger, 1998; Szabo & Ostlund, 1996).

**Newton-Schulz and polynomial Padé methods.** The earliest methods in the literature are polynomial iterations like (2). Several nearly simultaneous papers introduced the family of polynomial Padé iterations, comprising Newton-Schulz and its higher-degree analogues (Kovářík, 1970; Björck & Bowie, 1971; Higham, 1986; Leipnik, 1971). These higher-degree methods are also sometimes called "Newton-Schulz"; when doing so, we will specify the degree for clarity. In these methods, each iteration refines the current approximation $X_t$ by applying a low-degree odd matrix polynomial, where any odd monomial $x \mapsto x^{2q+1}$ is defined for rectangular matrices by the formula $X_t \mapsto X_t \left( X_t^\top X_t \right)^q$. Our Polar Express method also takes this form, though unlike Newton-Schulz, it changes the polynomial at each iteration.

The polynomials used in Padé methods are chosen to match the value and first few derivatives of $\mathrm{sign}(x)$ at the points $x = \pm 1$. For instance, the update rule of the third method in this family is defined by $p(x) = \frac{1}{16} \left( 35x - 35x^3 + 21x^5 - 5x^7 \right)$, which is the unique degree-7 polynomial satisfying $p(\pm 1) = \pm 1$ and $p'(\pm 1) = p''(\pm 1) = p'''(\pm 1) = 0$. These methods converge so long as all singular values of $X_0$ lie in $(0, 1]$, a condition guaranteed by the initialization of (2). Furthermore,

the order of convergence of the degree $2q+1$ method is $q+1$ (Björck & Bowie, 1971). In particular, the Newton-Schulz method ($q = 1$) converges quadratically.

**Newton's method and rational Padé.** In the numerical analysis literature, polynomial methods were succeeded by rational iterations like Newton's method (Higham, 1986), defined as follows[8]:

$$\boldsymbol{X}_0 = \boldsymbol{M} \qquad\qquad \boldsymbol{X}_{t+1} = \frac{1}{2}\left(\boldsymbol{X}_t + \boldsymbol{X}_t^{-\top}\right) \qquad (14)$$

Newton's method also converges quadratically. Like Newton-Schulz, it works because the rational function $r(x) = \frac{1}{2}(x + x^{-1})$ has a stable fixed point at 1; unlike for Newton-Schulz, this point is a global attractor for the whole positive real line. At first glance, Newton's method has nothing to do with the Padé iterations discussed above. However, after a change of variables $\boldsymbol{Y}_t = \boldsymbol{X}_t^{-1}$, it can be reinterpreted as $\boldsymbol{Y}_{t+1} = 2\boldsymbol{Y}_t(\boldsymbol{I} + \boldsymbol{Y}_t^\top \boldsymbol{Y}_t)^{-1}$, which is sometimes called inverse Newton. Observing that $r(x) = \frac{2x}{1+x^2}$ satisfies $r(\pm 1) = \pm 1$ and $r'(\pm 1) = 0$, we see that (inverse) Newton is also a Padé method, though a rational rather than polynomial one. In fact, given a odd degree $2q_n + 1$ for the numerator and an even degree $2q_d$ for the denominator, there is a unique rational function that matches the value and first $q_n + q_d$ derivatives of $\mathrm{sign}(x)$ at $x = \pm 1$. This directly yields a Padé method for computing $\mathrm{polar}(\boldsymbol{M})$ whose order of convergence is $q_n + q_d + 1$. For instance, $r(x) = \frac{3x+x^3}{1+3x^2}$ is called Halley's method, which converges cubically. When $q_d = 0$, we recover the polynomial Padé methods.

There are two main weakness of Newton's method and the Padé iterations: slow convergence in the initial phase and the need to compute explicit inverses. To accelerate initial convergence, Higham popularized the technique of rescaling the matrix after every Newton iteration (Higham, 1986). Intuitively, rescaling $\boldsymbol{X}_t$ so that $\sigma_{\max} = 1/\sigma_{\min}$ centers the spectrum around 1, where convergence is fastest. Several easily-computable choices of scaling factor exist to accomplish this approximately. Note that this rescaling scheme would fail for Newton-Schulz, which likewise suffers from slow initial convergence but which would diverge if $\sigma_{\max} \gg 1$.

Computing matrix inverses is difficult to parallelize and to implement stably in low precision arithmetic. However, a trick was developed for stably computing many rational methods *without* explicit inverses; QR decompositions can be used instead (Nakatsukasa et al., 2010; Zhang et al., 2007). Applying this trick to Halley's method and combining with a special rescaling scheme yields the QDWH (QR-based dynamically weighted Halley) method, which converges in just six iterations for any reasonably conditioned matrix (Nakatsukasa et al., 2010).

**Adaptive rational methods from optimal approximations.** A landmark 2016 paper introduced a new paradigm to design iterative methods for computing $\mathrm{polar}(\boldsymbol{M})$ (Nakatsukasa & Freund, 2016). The main insight is as follows. Padé methods choose the update rule to be an approximation to $\mathrm{sign}(x)$ of a given degree that is optimally accurate in the neighborhood of $x = 1$. Instead, we should choose the approximation to $\mathrm{sign}(x)$ that is optimal over an *interval* $[\ell, 1] \subset \mathbb{R}_{\geq 0}$ that contains the singular values. Moreover, after each step of the algorithm, the range of the singular values changes; therefore, we adapt the update rule at each iteration to match the new interval. When the range of the singular values is large, this approach ensures that the update rule shrinks it as quickly as possible. As the algorithm proceeds and the interval shrinks to a small neighborhood of 1, the update rule approaches that of a Padé method, maintaining the same high order of convergence as it has.

Within the class of odd rational functions whose numerators and denominators have degree $2q + 1$ and $2q$, respectively, an explicit formula for this optimal approximation to $\mathrm{sign}(x)$ on any interval $[\ell, 1]$ was found by Zolotarev. It was shown that these rationals have remarkable convergence properties for any $q$ (Nakatsukasa & Freund, 2016). For $q = 1$, this optimal approximation coincides exactly with the dynamically weighted Halley's method (QDWH) referenced above. For even faster convergence than QDWH, (Nakatsukasa & Freund, 2016) proposed the Zolo-pd method, which uses $q = 17$. Finally, these methods all admit the same QR-based implementation trick as QDWH.

**Adaptive polynomial methods.** In this paper, we adopt the paradigm of Zolo-pd (Nakatsukasa & Freund, 2016) but with polynomials rather than rationals of degree $(2q + 1, 2q)$. This choice

---

[8]Our description of Newton's method and other rational methods assumes square non-singular $\boldsymbol{M}$. Non-square problems can be reduced to the square case by an initial QR decomposition, but this is not an option for purely polynomial methods like ours.

avoids the need for QR factorizations, relying solely on GPU-friendly matrix-matrix multiplications in low-precision arithmetic. While this class of methods has not been fully developed in the numerical analysis literature, similar ideas have been rediscovered in different guises. In an unpublished manuscript that predates Zolo-pd, Chen & Chow (2014) describe a rescaling strategy for Newton-Schulz. Though motivated differently, their method is equivalent to ours for degree-3 polynomials (unlike our work, they do not consider general odd degree). They also observe numerical instability that prevents the method from converging to all the way to machine precision. Using the insights of Nakatsukasa & Higham (2012), they propose a simple mitigation for this issue that we adopt in Section 3.4. Our work gives the approach from Nakatsukasa & Higham (2012) a stronger theoretical foundation that connects to the paradigm of Zolo-pd. Concretely, we prove that choosing an optimal polynomial at each iteration leads to a composed polynomial that is *globally* optimal in the sense of (5).

Independently, a group of cryptographers developed a similar method for approximating the scalar function $\text{sign}(x)$ in the context of homomorphic encryption schemes (Lee et al., 2022). Their focus is mainly on tuning the analogues in their setting of the polynomial degree and number of iterations, whereas we focus on demonstrating optimality and efficiently constructing the update polynomials for degree 3 and 5. In addition, we consider matrix-valued inputs in low-precision arithmetic—not scalars in exact arithmetic—and we demonstrate our method's effectiveness within the `Muon` algorithm for training deep neural networks.

**Application within `Muon`.** The designers of `Muon` realized that, due to the extreme efficiency requirements and lax accuracy requirements of their setting, rational-based methods from the numerical analysis literature are inapplicable. However, polynomial-based iteration schemes can take full advantage of GPUs because they use only matrix-matrix products in half-precision arithmetic, not inverses or QR decompositions. The preference for speed over accuracy motivates methods that aim to quickly produce coarse approximations, even at the cost of asymptotic convergence. Examples include the proposals of Jordan (Jordan et al., 2024b) and You (Cesista et al., 2025), as discussed in Section 1.2. Like Chen & Chow (2014), Jordan found that convergence in the initial phase can be accelerated by choosing update rules that have a large derivative near zero, so as to increase the small singular values as much as possible at each iteration. You furthermore chose to use different update rules at each iteration, allowing extra flexibility to tune the trade-off between speed and accuracy. Both used degree-5 polynomials that were found through gradient descent on heuristic objective functions. These proposals were previously compared to Newton-Schultz[9], but never to Nakatsukasa & Higham (2012). We find that our method (which generalizes Nakatsukasa & Higham (2012)) outperforms them all.

Finally, we remark that concurrent work of Grishina, Smirnov, and Rakhuba also proposes an adaptive polynomial method that generalizes Nakatsukasa & Higham (2012) and applies it to accelerating `Muon` (Grishina et al., 2025). Like Nakatsukasa & Higham (2012), this work does not establish global optimality of the composed polynomial as we do in Section 3 or address finite precision considerations.

## C  PROOF OF THEOREM 3.1

The aim of this section is to prove Theorem 3.1. We begin with a result that provides a few essential properties for the the polynomial solving (6) when $T = 1$. This result is known as Chebyshev's theorem (Chebyshev, 1947) or the equioscillation theorem (Trefethen, 2020, Chapter 10).

**Lemma C.1.** Let $d = 2q + 1$ and $u, \ell > 0$. Consider the problem

$$\min_{p \in \mathbb{P}_d^{\text{odd}}} \max_{x \in [\ell, u]} |1 - p(x)|. \tag{15}$$

---

[9]Jordan et al. (2024b) actually compares to $2x - \frac{3}{2}x^3 + \frac{1}{2}x^5$, whereas the true degree-5 Newton-Schulz polynomial is $(15x - 10x^3 + 3x^5)/8$. However, the difference in performance is negligible for the first few iterations.

There exists a unique polynomial $p^\star \in \mathbb{P}_d^{\text{odd}}$ solving (15). Furthermore, $p^\star$ is the unique solution to the above problem if and only if there exist $q + 2$ distinct points $\{x_0, \ldots, x_{q+1}\} \subset [\ell, u]$ such that

$$1 - p^\star(x_i) = \eta(-1)^i \max_{x \in [\ell, u]} |1 - p^\star(x)|, \quad \text{for } i = 0, \ldots, q + 1,$$

for $\eta = 1$ or $\eta = -1$.

*Proof.* A discussion can be found in Eremenko & Yuditskii (2007). Here we include a formal proof for completeness.

By Chebyshev's Theorem (Achieser, 1992; Chebyshev, 1947; Cheney, 1966) it is sufficient to show that $\mathbb{P}_d^{\text{odd}}$ satisfies the Haar condition: any non-zero $p \in \mathbb{P}_d^{\text{odd}} = \text{span}\{x, \ldots, x^3, \ldots, x^{2q+1}\}$ can have at most $q$ roots in $[\ell, u]$.

Since $\deg(p) = d = 2q + 1$ we know that $p$ can have at most $2q + 1$ roots in $\mathbb{R}$. However, since $p(0) = 0$ and $p(x) = -p(-x)$ we know that $p$ has one root at zero, and the remaining roots come in symmetric pairs $(x, -x)$. Because of this, $p$ can have at most $q$ roots in the positive orthant, and thus it can have at most $q$ roots in $[\ell, u] \subset (0, \infty)$. Hence, $\mathbb{P}_d^{\text{odd}}$ satisfies the Haar condition, which yields the desired result.

$\square$

The proof of Theorem 3.1 will be by induction on $T$. We begin by establishing the base case, $T = 1$, which is handled by the following result.

**Lemma C.2.** Let $u, \ell > 0$ and define

$$p^\star := \arg\min_{p \in \mathbb{P}_d^*} \max_{x \in [\ell, u]} |1 - p(x)|.$$

Then

$$p^\star(\ell) = \min_{x \in [\ell, u]} p^\star(x), \quad \max_{x \in [\ell, u]} p^\star(x) = 2 - p^\star(\ell), \text{ and } \max_{x \in [\ell, u]} |1 - p^\star(x)| = 1 - p^\star(\ell).$$

*Proof.* Throughout the proof we assume $d = 2q + 1$. We begin with proving

$$p^\star(\ell) = \min_{x \in [\ell, u]} p^\star(x).$$

Consider the polynomial $e(x) := 1 - p^\star(x)$. The proof will contain three steps. We first rule out the trivial case that $p^\star \neq 0$, since $p(x) = \frac{2}{\ell + u} x$ would then be a better approximation. Hence, $p^\star$ cannot be the zero polynomial.

*Step 1: $e(x)$ has exactly $q$ stationary points inside the open interval $(\ell, u)$.*

Note that $e(x)$ has at most $2q$ stationary points in $\mathbb{R}$, since its derivative $e'(x)$ is a polynomial of degree $2q$. Furthermore, since $p^\star$ is odd, we have that $e'(x) = -p'(x)$ is even of degree $2q$, and thus can have at most $q$ stationary points contained in $(0, +\infty)$. Hence, there can be at *most* $q$ stationary points of $e(x)$ inside the interval $[\ell, u]$.

By Lemma C.1 there are $q + 2$ points $x_0, \ldots, x_{q+1} \in [\ell, u]$ where $e(x)$ is maximized or minimized in $[\ell, u]$. These points are either stationary points or they are endpoints of the interval $[\ell, u]$. Let $n_{\text{ext}}$ be the number of stationary points and $n_{\text{stat}}$ be the number of endpoints in the set $\{x_0, \ldots, x_{q+1}\}$. Since a point can be both a stationary point and an endpoint we have $q + 2 \leq n_{\text{end}} + n_{\text{stat}}$. However, $n_{\text{end}} \leq 2$ and $n_{\text{stat}} \leq q$, which follows from the previous paragraph where we showed that there are at most $q$ stationary points of $e(x)$ in $[\ell, u]$. So $n_{\text{end}} + n_{\text{stat}} \leq q + 2$, and consequently we must have $n_{\text{end}} = 2$ and $n_{\text{stat}} = q$, as required.

*Step 2: $x = \ell$ is a maximum of $e(x)$ on the interval $[\ell, u]$*

By Lemma C.1 and the discussion from Step 1, we know that $|e(x)|$ is maximized at $q + 2$ points inside $[\ell, u]$ and $q$ of these points are contained inside the open interval $(\ell, u)$. Hence, $x = \ell$ must

either be a maximum or a minimum of $e(x)$. We will show that $x = \ell$ must be a maximum by contradiction.

Suppose $x = \ell$ was a minimum of $e(x)$ on $[\ell, u]$. First note that $p^\star$ is trivially non-negative on $[\ell, u]$, or else $p(x) = 0$ would be a better polynomial. Hence, since $p^\star(0) = 0$ we must have $p^{*'}(\delta) > 0$ for some $\delta \in [0, \ell]$, or else the zero polynomial $p(x) = 0$ would be a better approximation. Hence, for some $\delta \in [0, \ell]$ we have $e'(\delta) < 0$.

We must also have $e'(\ell) \geq 0$ or else $x = \ell$ is not a minimum of $e(x)$. Since $e'(\delta) < 0$ for some $\delta \in [0, \ell]$ and $e'(\ell) \geq 0$, by the intermediate value theorem there exists a point $x^* \in [0, \ell]$ such that $e'(x^*) = 0$. However, by the discussion above we know that all stationary points of $e$ are contained inside the open interval $(\ell, u)$. Hence, $x = \ell$ cannot be a minimum of $e(x)$ on $[\ell, u]$. However, by Step 1 we know that the endpoints of $[\ell, u]$ must be either minima or maxima of $e(x)$. Hence, $x = \ell$ is a maximum of $e(x)$ on $[\ell, u]$.

*Step 3: Obtaining the desired equalities*

Since $e(x)$ has a maximum in $[\ell, u]$ at $x = \ell$, we have $p^\star(\ell) = \min_{x \in [\ell, u]} p^\star(x)$. The other two equalities are immediate consequences of the equioscillation property of $p^\star$ Lemma C.1 and that $x = \ell$ is a minimum of $p^\star$ over the set $[\ell, u]$. □

With the above-mentioned result in hand, we are ready to prove Theorem 3.1.

---

**Theorem 3.1.** Let $d$ be odd and define $\ell_1 = \ell$ and $u_1 = u$. For $t = 1, \ldots, T$ define

$$p_t = \underset{p \in \mathbb{P}_d^{\mathrm{odd}}}{\arg\min} \max_{x \in [\ell_t, u_t]} |1 - p(x)|, \quad \ell_{t+1} = \min_{x \in [\ell_t, u_t]} p_t(x), \quad u_{t+1} = \max_{x \in [\ell_t, u_t]} p_t(x) \quad (8)$$

The resulting composition $p^\star := p_T \circ p_{T-1} \circ \cdots \circ p_1$ is optimal and the error is given by:

$$\max_{x \in [\ell, u]} |1 - p^\star(x)| = \min_{\substack{p = p_T \circ p_{T-1} \circ \cdots \circ p_1 \\ p_t \in \mathbb{P}_d^{\mathrm{odd}}}} \max_{x \in [\ell, u]} |1 - p(x)| = 1 - \ell_{T+1}. \quad (9)$$

Furthermore the new error, lower and upper bounds can be computed through

$$\ell_{t+1} = p_t(\ell_t), \quad u_{t+1} = 2 - \ell_{t+1}, \quad \text{and} \quad \max_{x \in [\ell_t, u_t]} |1 - p_t(x)| = 1 - \ell_{t+1}. \quad (10)$$

---

*Proof.* The proof of (10) is an immediate consequence of Lemma C.2, since for each $t = 1, \ldots, T$, $p_t$ is the optimal approximation in $\mathbb{P}_d^{\mathrm{odd}}$ to $x \mapsto 1$.

We now proceed with the proof of (9), which will be by induction. The proof for $T = 1$ is an immediate consequence of Lemma C.2 and we also have $p^\star(\ell) = \ell_2$ by (10). Now suppose the result is true for all $t \leq T - 1$. Thus

$$g(x) := p_{T-1} \circ \cdots \circ p_1(x)$$

is the optimal solution of (9) for $T - 1$. For $t = 1, \ldots, T - 1$, note that the image of $p_t$ on $[\ell_t, u_t]$ is exactly $[\ell_{t+1}, u_{t+1}]$ by Lemma C.2. Hence, the image of $g$ on $[\ell, u]$ is $[\ell_T, u_T]$. Furthermore, by Lemma C.2 we also have $g(\ell) = \ell_T$. Pick any $f$ such that $f \neq g$ and

$$f = \widetilde{p}_{T-1} \circ \cdots \circ \widetilde{p}_1,$$

for some $\widetilde{p}_1, \ldots, \widetilde{p}_{T-1} \in \mathbb{P}_d^{\mathrm{odd}}$. Let the image of $f$ on $[\ell, u]$ be $[a, b]$. We will prove that $\frac{a}{b} \leq \frac{\ell_T}{u_T}$ by contradiction.

Suppose $\frac{a}{b} > \frac{\ell_T}{u_T}$. Define $c = \frac{2}{a+b}$. Then, the image of the scaled function $cf$ on $[\ell, u]$ is $[ca, cb]$ and $cf$ satisfies

$$\max_{x \in [\ell, u]} |1 - cf(x)| = \max\{1 - ca, cb - 1\} = \frac{b - a}{a + b}.$$

Recall by our inductive hypothesis, we have $\max_{x\in[\ell,u]}|1-g(x)| = 1 - \ell_T = u_T - 1$ where the second equality holds by (10). It follows that

$$\frac{a}{b} > \frac{\ell_T}{u_T}$$

$$\Leftrightarrow \frac{a}{b} > \frac{\ell_T}{2-\ell_T}$$

$$\Leftrightarrow \ell_T < \frac{2a}{a+b}$$

$$\Leftrightarrow 1-\ell_T > \frac{b-a}{a+b}$$

$$\Leftrightarrow \max_{x\in[\ell,u]}|1-g(x)| > \max_{x\in[\ell,u]}|1-cf(x)|,$$

which leads to a contradiction to our inductive hypothesis that $g$ is optimal. Hence, we must have $\frac{a}{b} \le \frac{\ell_T}{u_T}$.

Consequently, using that $\frac{a}{b} \le \frac{\ell_T}{u_T}$, we will show for any $\widetilde{p}_T \in \mathbb{P}_d^{\mathrm{odd}}$ and for any $f = \widetilde{p}_{T-1} \circ \cdots \circ \widetilde{p}_1$, that $\widetilde{p}_T \circ f$ cannot be a better approximation than $p_T \circ g$. In particular, we have

$$\max_{x\in[\ell,u]}|1-\widetilde{p}_T(f(x))| \ge \min_{p\in\mathbb{P}_d^*}\max_{x\in[\ell,u]}|1-p(f(x))|$$

$$= \min_{p\in\mathbb{P}_d^*}\max_{x\in[a,b]}|1-p(x)|$$

$$= \min_{p\in\mathbb{P}_d^*}\max_{x\in[a/b,1]}|1-p(x)|$$

$$\ge \min_{p\in\mathbb{P}_d^*}\max_{x\in[\ell_T/u_T,1]}|1-p(x)|$$

$$= \min_{p\in\mathbb{P}_d^*}\max_{x\in[\ell_T,u_T]}|1-p(x)|$$

$$= \min_{p\in\mathbb{P}_d^*}\max_{x\in[\ell,u]}|1-p(g(x))|$$

$$= \max_{x\in[\ell_T,u_T]}|1-p_T(g(x))| = 1 - p_T(\ell_T) = 1 - \ell_{T+1},$$

where the second and third equality follow by changing variables $y = x/b$ so that

$$\min_{p\in\mathbb{P}_d^*}\max_{x\in[a,b]}|1-p(x)| = \min_{p\in\mathbb{P}_d^*}\max_{y\in[a/b,1]}|1-p(by)| = \min_{p\in\mathbb{P}_d^*}\max_{y\in[a/b,1]}|1-p(y)|$$

and this last equality follows because the space $\mathbb{P}_d^*$ is invariant under input rescaling; that is, for any $b \ne 0$, the map $x \mapsto bx$ preserves the space $\mathrm{span}\{x, x^3, \ldots, x^d\}$. This concludes the proof. $\square$

## D  PROOF OF THEOREM 3.3

In this section we provide the proof of the convergence guarantee stated in Theorem 3.3.

**Theorem 3.3.** Let $M$ be a matrix normalized so that $\sigma(M) \subset [\ell, 1]$. Let $X_T = p^\star(M)$, where $p^\star$ is the polynomial from Theorem 3.1 with $d = 2q + 1$. Then, we have

$$\| \mathrm{polar}(M) - X_T \|_2 \le |1 - \ell^2|^{(q+1)^T}. \tag{12}$$

Hence, for $d = 3$ and $d = 5$ the method converges quadratically and cubically, respectively.

*Proof.* Define

$$p^\star = \underset{\substack{p = p_T \circ p_{T-1} \circ \cdots \circ p_1 \\ p_t \in \mathbb{P}_d^*}}{\arg\min} \max_{x\in[\ell,u]} |1 - p(x)|.$$

Then Algorithm 1 returns $\boldsymbol{X}_T = p^\star(\boldsymbol{M})$. Let $h \in \mathbb{P}_q$ be the $[q/0]$ Padé-approximant to $(1-x)^{-1/2}$ (Kenney & Laub, 1991, Section 3) and define $p(x) = xh(1-x^2) \in \mathbb{P}_d^{\mathrm{odd}}$. Define $f = p \circ \cdots \circ p$ as the composition of $p$ with itself $T$ times. Then, by Theorem 3.1, (Kenney & Laub, 1991, Theorem 3.1), and $f(x) \geq 0$ for $x \geq 0$ we have

$$
\begin{aligned}
\|\operatorname{sign}(\boldsymbol{M}) - \boldsymbol{X}_T\|_2 &\leq \max_{x \in [\ell,1]} |1 - p^\star(x)| \\
&\leq \max_{x \in [\ell,1]} |1 - f(x)| \\
&\leq \max_{x \in [\ell,1]} \left[ \frac{|1 - x^2|^{(d+1)^T}}{1 + f(x)} \right] \\
&\leq |1 - \ell^2|^{(d+1)^T},
\end{aligned}
$$

as required. $\qquad\square$

## E  PROOF OF EQUIVALENCE BETWEEN (5) AND (6)

In this section we provide a proof for the equivalence between (5) and (6). It is sufficient to show that for any fixed polynomial $p$ we have

$$
\varepsilon_1 := \max_{\substack{\boldsymbol{M} \in \mathbb{R}^{m \times n} \\ \sigma(\boldsymbol{M}) \subset [\ell,u]}} \|\operatorname{polar}(\boldsymbol{M}) - p(\boldsymbol{M})\|_2 = \max_{x \in [\ell,u]} |1 - p(x)| := \varepsilon_2.
$$

For any fixed $\boldsymbol{M}$, by the unitary invariance of the spectral norm we immediately have

$$
\|\operatorname{polar}(\boldsymbol{M}) - p(\boldsymbol{M})\|_2 = \max_{\sigma_i \in \sigma(\boldsymbol{M})} |1 - p(\sigma_i)| \leq \max_{x \in [\ell,u]} |1 - p(x)|.
$$

Consequently, $\varepsilon_1 \leq \varepsilon_2$.

Suppose that $x^* \in [\ell,u]$ is chosen so that $|1 - p(x^*)| = \max_{x \in [\ell,u]} |1 - p(x)|$. Without loss of generality, assume $m \geq n$. Letting $\boldsymbol{M} = x^* \boldsymbol{U} \boldsymbol{V}^\mathsf{T}$, for any matrix $\boldsymbol{U} \in \mathbb{R}^{m \times n}$ and $\boldsymbol{V} \in \mathbb{R}^{n \times n}$ with orthonormal columns, and noting $\operatorname{polar}(\boldsymbol{M}) = \boldsymbol{U} \boldsymbol{V}^\mathsf{T}$ yields

$$
\begin{aligned}
\varepsilon_1 &\geq \|\operatorname{polar}(\boldsymbol{M}) - p(\boldsymbol{M})\|_2 \\
&= \|\boldsymbol{I}_n - p(x^*) \boldsymbol{I}_n\|_2 \\
&= |1 - p(x^*)| \\
&= \max_{x \in [\ell,u]} |1 - p(x)| = \varepsilon_2
\end{aligned}
$$

Consequently, $\varepsilon_1 \geq \varepsilon_2$. Hence, $\varepsilon_1 = \varepsilon_2$, as desired.

## F  REMEZ ALGORITHM

In this section, we show in detail how to solve (13). By Theorem 3.1, these solutions give the update rule for a single step of Polar Express. We give a closed form solution for $d = 3$. We then describe how the Remez algorithm (Pachón & Trefethen, 2009; Parks & McClellan, 1972) can be used to approximate $p_t$ for arbitrary $d$. We then present Algorithm 2, a simplified version of Remez for solving (13) with $d = 5$. Recall (13):

$$
\arg\min_{p \in \mathbb{P}_d^{\mathrm{odd}}} \max_{x \in [\ell,u]} |1 - p(x)|
$$

We begin with the case when $d = 3$. We seek a polynomial of the form $p(x) = ax + bx^3$. The Equioscillation Theorem (Lemma C.1) stipulates that $p$ must have an equioscillating set of size 3. For $p$ to achieve its maximum error at a point $x$, $x$ must be a local extremum of $p(x) - 1$ on the interval $[\ell,u]$. Thus, for $x$ to be eligible for membership in the equioscillating set, it must either be a true local extremum of $p(x) - 1$ that happens to lie in $[\ell,u]$, or else one of the endpoints $\ell, u$. However, because $p$ is an odd cubic, it has at most one true local extremum on $\mathbb{R}_{\geq 0}$. Thus, to build

an equioscillating set of three points, we must include $p$'s unique positive local extremum *and* both endpoints. This local extremum of $p$ occurs at $\sqrt{\frac{-a}{3b}}$. Therefore, we seek $a, b$ such that

$$p(\ell) = 1 - E, \qquad p\left(\sqrt{\frac{-a}{3b}}\right) = 1 + E, \qquad p(u) = 1 - E \qquad (16)$$

for some $E$. This is a system of three equations in three variables. The solution $p(x) = ax + bx^3$ is most easily expressed as follows. Let $p_{\mathrm{NS}}(x) = \frac{3}{2}x - \frac{1}{2}x^3$. Then

$$p(x) = \beta p_{\mathrm{NS}}(\alpha x), \quad \text{where } \alpha = \sqrt{\frac{3}{u^2 + \ell u + \ell^2}} \quad \text{and} \quad \beta = \frac{4}{2 + \ell u(\ell + u)\alpha^3}. \qquad (17)$$

One can verify that this polynomial satisfies the equioscillation condition of (16), with $\sqrt{\frac{-a}{3b}} = \frac{1}{\alpha}$ and $E = \beta - 1$. Therefore, it must necessarily be the optimal approximation from $\mathbb{P}_3^{\mathrm{odd}}$. Note that for $u = 1$, $x \mapsto p_{\mathrm{NS}}(\alpha x)$ is the same polynomial derived in Chen & Chow (2014).

Unfortunately, for larger $d$, finding closed form expressions for optimal approximations from $\mathbb{P}_d^{\mathrm{odd}}$ becomes challenging, and we know of no closed form solution. However, we can approximate the optimal polynomial using the Remez algorithm. Let $d = 2q + 1$. Again recalling Lemma C.1, the optimal polynomial must satisfy the equioscillation property at a set of $q + 2$ points, as in (16). The Remez algorithm finds the equioscillation points $A = \{x_0, \ldots, x_{q+1}\}$ from Lemma C.1 by iteratively refining a sequence of trial points $A^{(k)} = \{x_0^{(k)}, \ldots, x_{q+1}^{(k)}\}$ so that $A^{(k)}$ converges to $A$. From the sequence of trial points $A^{(k)}$ the algorithm also finds a sequence of polynomials $p^{(k)}$ so that $p^{(k)}$ converges to the optimal polynomial. The convergence is very fast, and usually 10 iterations is sufficient to converge to the optimal polynomial up to double precision machine epsilon (Pachón & Trefethen, 2009). More commonly, the Remez algorithm is used to find optimal polynomial approximations to general continuous functions where $d \approx 100$ or even $d \approx 1000$. However, because the polynomial we build to approximate $\mathrm{sign}(x)$ is a composition of polynomials, each of which has a low degree, in our setting the degree $d$ is small, usually $d = 5$. For $d = 5$ the Remez algorithm simplifies significantly. We now describe this simplified algorithm.

We first choose an initial set of trial points $A^{(1)}$, which ideally should come close to satisfying the equioscillation property. From Lemma C.1, the unique optimal approximation $p^\star \in \mathbb{P}_5^{\mathrm{odd}}$ satisfies the equioscillation property at four points in $[\ell, u]$. Since the function we wish to approximate is constant, the equioscillation points must be extrema of $p^\star$ on $[\ell, u]$. Because $p^\star$ is a odd quintic, it can have at most two local extrema on the positive real line, and thus at most two local extrema on $[\ell, u]$. The other two equioscillation points must therefore be the endpoints $\ell$ and $u$. Since we know that $\ell$ and $u$ must be equioscillation points we always set $x_0^{(k)} = \ell$ and $x_3^{(k)} = u$ for all $k$. We initialize $x_1^{(1)}$ and $x_2^{(1)}$ to $\frac{3}{4}\ell + \frac{1}{4}u$ and $\frac{1}{4}\ell + \frac{3}{4}u$, since we observe that as $\ell \to u$ these are approximately the other two equioscillation points.

We now show how to refine a candidate set of trial points $A^{(k)}$ to produce $A^{(k+1)}$ as well as an approximately equioscillating polynomial $p_k$. For any fixed set of trial points $\{\ell, x_1^{(k)}, x_2^{(k)}, u\}$, we can find a degree-5 odd polynomial $p_k(x) = a_k x + b_k x^3 + c_k x^5$ that satisfies

$$p_k(\ell) = 1 - E_k, \quad p_k(x_1^{(k)}) = 1 + E_k, \quad p_k(x_2^{(k)}) = 1 - E_k, \quad p_k(u) = 1 + E_k \qquad (18)$$

for some $E_k$ by solving a linear system in $a_k, b_k, c_k$ and $E_k$. This can be rewritten as follows:

$$\begin{bmatrix} \ell & \ell^3 & \ell^5 & 1 \\ x_1^{(k)} & (x_1^{(k)})^3 & (x_1^{(k)})^5 & -1 \\ x_2^{(k)} & (x_2^{(k)})^3 & (x_2^{(k)})^5 & 1 \\ u & u^3 & u^5 & -1 \end{bmatrix} \begin{bmatrix} a_k \\ b_k \\ c_k \\ E_k \end{bmatrix} = \begin{bmatrix} 1 \\ 1 \\ 1 \\ 1 \end{bmatrix}. \qquad (19)$$

If $A^{(k)}$ were the extrema of the error function $e_k(x) = 1 - p_k(x)$ on $[\ell, u]$, then they would be an equioscillating set for $p_k$, and $p_k$ would be the solution. Therefore, to refine $A^{(k)}$, we find the extrema of $e_k(x) = 1 - p_k(x)$. These can occur at $\ell, u$ and the roots of $e_k'(x)$. Setting $e_k'(x) = 0$

yields the quartic equation $5c_k x^4 + 3b_k x^2 + a_k = 0$, whose two solutions are given explicitly by the *quadratic* formula after the substitution $y = x^2$. We set $x_1^{(k+1)}$ and $x_2^{(k+1)}$ to be the solutions to this equation and let $A^{(k+1)} = \{\ell, x_1^{(k+1)}, x_2^{(k+1)}, u\}$. We repeat the procedure until $|E_k| := \max_{x \in [\ell,u]} |1 - p_k(x)| \approx \max_{x \in [\ell,u]} |1 - p_{k+1}(x)| =: |E_{k+1}|$.

We note that the matrix appearing in (19) is a Vandermonde matrix. Vandermonde matrices become notoriously ill-conditioned as the degree grows large (Golub & Van Loan, 2013, Section 4.6). However, since in our setting we choose $d$ to be small, there is no ill-conditioning due to large degrees. Instead, we observe ill-conditioning when $\ell \approx u$. However, as $\ell/u \to 1$ the optimal polynomial will converge to the polynomial $\frac{x/u}{8} \left(15 - 10(x/u)^2 + 3(x/u)^4\right)$, which can be verified by noting that as $\ell/u \to 1$ all equioscillation points $x_0, x_1, x_2, x_3$ must converge to $u$. For general $d = 2q + 1$, the polynomial will converge to $(x/\ell)h(1 - (x/\ell)^2)$ where $h \in \mathbb{P}_q$ is the $[q/0]$ Padé approximant to $(1 - x)^{1/2}$ (Kenney & Laub, 1991). In fact, this polynomial is extremely close to the optimal polynomial for sufficiently large $\ell$. To see this, let $p^\star$ be the optimal approximation from $\mathbb{P}_5^{\text{odd}}$ and let $p(x) = \frac{x/u}{8} \left(15 - 10(x/u)^2 + 3(x/u)^4\right)$. Then,

$$\max_{x \in [\ell,u]} |p^\star(x) - p(x)| \leq \max_{x \in [\ell,u]} |1 - p(x)| + \max_{x \in [\ell,u]} |1 - p^\star(x)|$$
$$\leq 2 \max_{x \in [\ell,u]} |1 - p(x)|$$
$$\leq 2 \left(1 - \ell/u\right)^3.$$

where we invoked (Kenney & Laub, 1991, Theorem 3.1) and the fact that $p^\star$ is the optimal approximation to $x \mapsto 1$ from $\mathbb{P}_5^{\text{odd}}$. Hence, when $\ell/u \geq 1 - \epsilon_d^{1/3}$, where $\epsilon_{\text{double}} \approx 1.1 \times 10^{-16}$ is the double precision machine epsilon, then $|p^\star(x) - p(x)| \leq 2\epsilon_{\text{double}}$. In other words, up to double precision machine epsilon, $p^\star$ is equal to $p$. Therefore, whenever $\ell/u \geq 1 - \epsilon_{\text{double}}^{1/3}$ the algorithm simply returns the Padé approximant (that is, the scaled Newton-Schulz polynomial).

The full algorithm is given in Algorithm 2. In our experiments, we never observed Algorithm 2 taking more than five iterations to converge. This algorithm is implemented in full in Implementation 2.

---

**Algorithm 2** Remez algorithm (degree 5 approximation for $\text{sign}(x)$)

---

**input:** interval $[\ell, u]$ for $u > \ell > 0$.
**output:** Approximation $p \in \mathbb{P}_5^{\text{odd}}$ to $p^\star = \arg\min_{p \in \mathbb{P}_5^{\text{odd}}} \max_{x \in [\ell,u]} |1 - p(x)|$.

   **define** $\epsilon_{\text{double}} = 1.11 \times 10^{-16}$
   **if** $\ell/u \geq 1 - \epsilon_{\text{double}}^{1/3}$ **then**
      Return $p(x) = \frac{x/u}{8} \left(15 - 10(x/u)^2 + 3(x/u)^4\right)$
   **end if**
   $x_1^{(1)} = \frac{3}{4}\ell + \frac{1}{4}u, \quad x_2^{(1)} = \frac{1}{4}\ell + \frac{3}{4}u.$
   $E_0 = \infty, \quad E_{-1} = -\infty$
   $k \leftarrow 0$
   **while** $||E_k| - |E_{k-1}|| > \epsilon_{\text{double}}$ **do**
      $k \leftarrow k + 1$

$$\begin{bmatrix} a_k \\ b_k \\ c_k \\ E_k \end{bmatrix} = \begin{bmatrix} \ell & \ell^3 & \ell^5 & 1 \\ x_1^{(k)} & (x_1^{(k)})^3 & (x_1^{(k)})^5 & -1 \\ x_2^{(k)} & (x_2^{(k)})^3 & (x_2^{(1)})^5 & 1 \\ u & u^3 & u^5 & -1 \end{bmatrix}^{-1} \begin{bmatrix} 1 \\ 1 \\ 1 \\ 1 \end{bmatrix}$$

$$x_1^{(k+1)} = \sqrt{\frac{-3b_k - \sqrt{9b_k^2 - 20a_k c_k}}{10c_k}}, \quad x_2^{(k+1)} = \sqrt{\frac{-3b_k + \sqrt{9b_k^2 - 20a_k c_k}}{10c_k}}$$

   **end while**
   Return $p(x) = a_k x + b_k x^3 + c_k x^5$

---

## G  FINITE PRECISION CONSIDERATIONS

As highlighted in Section 3.4, one must take care to implement `Polar Express` in finite precision. In this section we outline modifications to our method to ensure stability in finite precision arithmetic.

The first issue arises when numerical round-off creates singular values that are slightly larger than our current upper bound $u_t$. Our optimal polynomials converge only when the singular values of $X_t$ are less than $u_t$. In some cases we have

$$p_t(u_t + \epsilon) > u_{t+1} + \epsilon,$$

so over many iterations, a singular value that is slightly larger than $u_t$ large could grow to $\infty$ instead of converging to 1.

To fix this issue, we simply replace each polynomial $x \mapsto p_t(x)$ by $x \mapsto p_t(x/1.01)$. This safety factor corrects for round-off errors in previous iterations while only slightly changing the behavior of the polynomial on the interval $[\ell_t, u_t]$, though it does cause the singular values to converge to 0.999998 instead of to 1. To correct for this, the safety factor can be omitted in the final iteration. This fix is reflected in line 5 of Algorithm 1.

The second issue was identified in Nakatsukasa & Higham (2012) and addressed in the context of polynomial iterations by Chen & Chow (2014). In general, iterative methods for $\text{polar}(M)$ aim to increase each singular value relative to the largest singular value; while $\sigma_{\min}(X_0) \ll \sigma_{\max}(X_0)$, after enough iterations, $\sigma_{\min}(X_t) \approx \sigma_{\max}(X_t) \approx 1$. However, the convergence of each singular value to $\sigma_{\max}$ may not be monotonic. Over the domain $[\ell_t, u_t]$, our optimal polynomial $p_t$ oscillates repeatedly between $\ell_{t+1}$ and $u_{t+1}$, so some singular values that are near $u_t$ may get mapped down to $\ell_{t+1}$. It so happens that this non-monotonicity—even at a single iteration—can cause loss of precision. That is, problems occur if

$$\frac{p_t(\sigma_i)}{\sigma_i} \ll \frac{\max\limits_{x \in [\sigma_{\min}, \sigma_{\max}]} p_t(x)}{\sigma_{\max}},$$

where $0 \le \sigma_{\min} \le \sigma_i \le \sigma_{\max}$ are singular values of $X_t$ (Nakatsukasa & Higham, 2012). In the extreme case $p_t(\sigma_i) < 0$, the $i$th singular vector will change sign, causing the method to converge to the polar factor of the wrong matrix. Unlike Newton-Schulz, unscaled Newton, or QDWH, our method is affected by this loss of precision.

To mitigate this issue, Chen & Chow (2014) propose modifying their update polynomials to enforce a lower bound on the ratio $\frac{p_t(\sigma_i)}{\sigma_i}$. This issue only occurs when $\ell_t \ll u_t$; as $\ell_t \to u_t$, our optimal polynomial approaches the Padé approximant and so $\frac{p_t(x)}{x} \ge 1$ for all $x \in [0, u_t]$. We could fully solve the problem by using the Padé approximant instead of our optimal polynomial, but this would significantly slow down convergence. Instead we compromise. When $\ell_t \ge u_t/10$, we find that $\frac{p_t(x)}{x} \ge 0.236$. Therefore, whenever $\ell_t < u_t/10$ we select the update rule as though $\ell_t = u_t/10$. This change slows convergence, but only very slightly. (The choice of 10 is somewhat arbitrary. In Implementation 2, we use a different factor.) This fix is reflected in line 4 of Algorithm 1.

The third change is copied from the original `Muon` implementation: normalize $M$ by $\|M\|_F + 10^{-2}$ instead of by $\|M\|_F$. As before, we set $u_1 = 1$. This fix is reflected in line 10 of Algorithm 1.

## H  ADDITIONAL EXPERIMENTAL RESULTS

In this section, we present additional experimental results.

### H.1  CONVERGENCE OF POLAR EXPRESS AND ITS IMPACT ON MUON

**Convergence in Frobenius Norm**    In Figure 8, we plot the convergence of `Polar Express` and three baselines as measured in the Frobenius norm. We also plot convergence in cosine similarity, which is defined with respect to the Frobenius inner product $\langle A, B \rangle = \text{Tr}(A^\top B)$. Formally, the cosine similarity between $A$ and $B$ is defined as $\frac{\langle A, B \rangle_F}{\|A\|_F \|B\|_F}$. We use gradients of GPT-2 layers as test matrices. While `Polar Express` is designed to minimize the spectral norm error, convergence in the Frobenius norm is similar (compare with Figure 3).

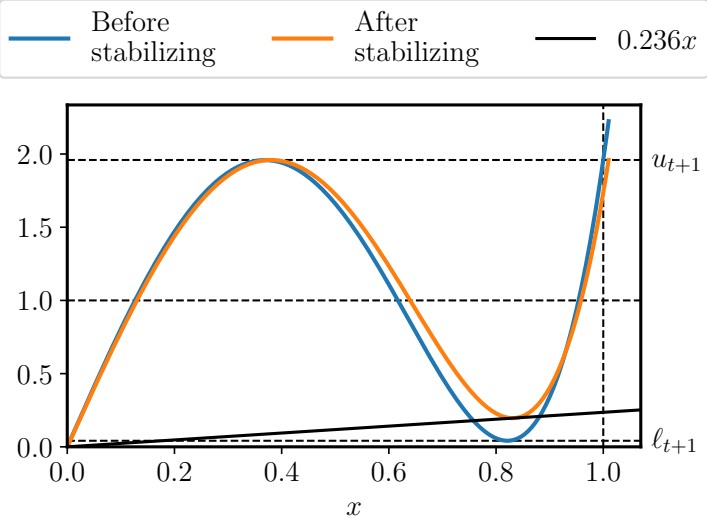

Figure 7: Effects of stabilizing the update rules with a safety factor and cushioning, as described in Appendix G. The blue curve is the optimal degree-5 polynomial for the interval $[0.005, 1]$. It is has numerical issues because it maps singular values near $0.8$ down to almost zero and maps $1 + \epsilon$ to $\approx u_{t+1} + 25\epsilon$. The stabilized version is better because it ensures $\frac{p_t(x)}{x} \geq 0.236$ and maps all $x \leq 1.01$ to at most $u_{t+1}$.

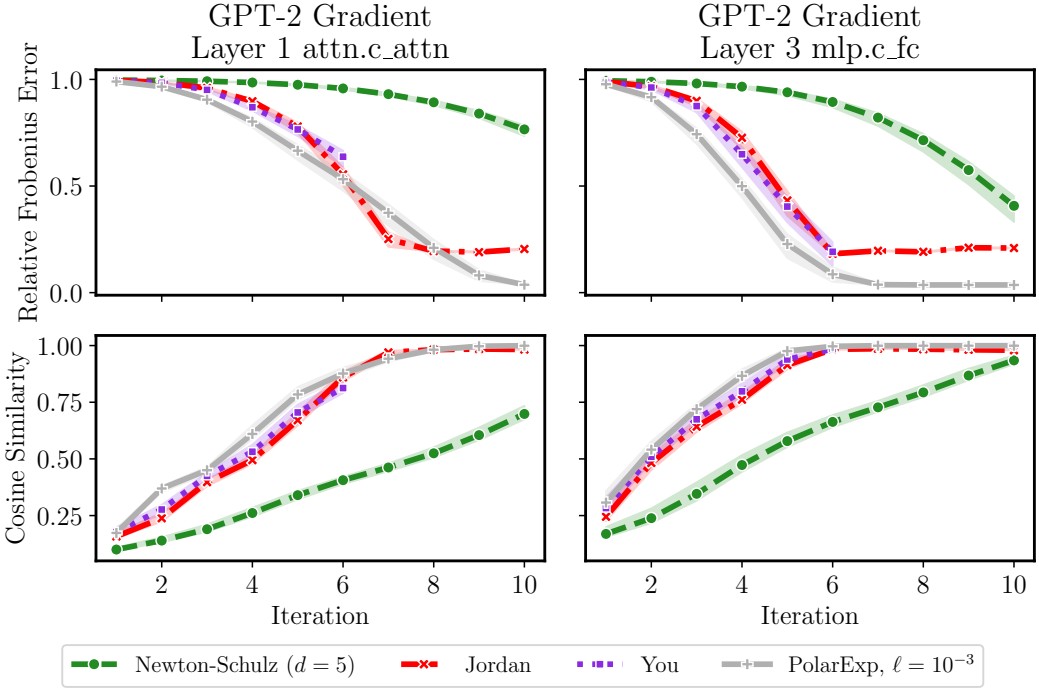

Figure 8: Convergence of degree-5 polynomial methods measured in Frobenius norm and cosine similarity. Test matrices are gradients of two layers of a randomly-initialized GPT-2 model on a batch of language modeling data. Polar Express outperforms other methods.

**(In)sensitivity of `Muon` to Small Singular Values**   Figure 5 shows that using more than five or six iterations of `Polar Express` does not improve the performance of `Muon`. However, Figures 3 and 8 show that five iterations is not enough for `Polar Express` or any other method to converge. In practice, `Polar Express` is taking steps in directions that are meaningfully different from the exact $\mathrm{polar}(\boldsymbol{M})$ (as computed by an SVD), but still converging equally fast. One possible explanation for this observation is that `Muon` may not be sensitive to the convergence of small singular values of $\boldsymbol{M}$. Intuitively, the singular vectors associated with these small singular values correspond to directions which have little effect on the output of the neural network; they may signify little more than noise in the stochastic gradients.

We now conduct an experiment to test this hypothesis. We compare three ways that a `Muon`-like optimizer could handle the small singular values. Assume $\boldsymbol{M}$ has full rank, and partition the singular value decomposition of $\boldsymbol{M}$ into two parts

$$\boldsymbol{M} = \boldsymbol{U}\boldsymbol{\Sigma}\boldsymbol{V}^\top = [\boldsymbol{U}_1 \quad \boldsymbol{U}_2] \begin{bmatrix} \boldsymbol{\Sigma}_1 & \\ & \boldsymbol{\Sigma}_2 \end{bmatrix} [\boldsymbol{V}_1 \quad \boldsymbol{V}_2]^\top = \boldsymbol{U}_1\boldsymbol{\Sigma}_1\boldsymbol{V}_1^\top + \boldsymbol{U}_2\boldsymbol{\Sigma}_2\boldsymbol{V}_2^\top \tag{20}$$

where $\boldsymbol{\Sigma}_1$ contains the singular values larger than some threshold $\gamma\sigma_{\max}$ and $\boldsymbol{\Sigma}_2$ contains those smaller than $\gamma\sigma_{\max}$, where $\sigma_{\max}$ is the largest singular value of $\boldsymbol{M}$. Recall that

$$\mathrm{polar}(\boldsymbol{M}) := \boldsymbol{U}\boldsymbol{V}^\top = \boldsymbol{U}_1\boldsymbol{V}_1^\top + \boldsymbol{U}_2\boldsymbol{V}_2^\top \tag{21}$$

is obtained by mapping each singular value of $\boldsymbol{M}$ to 1. We define the truncated polar factor by mapping the larger singular values to 1 and the smaller singular values to 0:

$$\mathrm{polar}_\gamma(\boldsymbol{M}) := \boldsymbol{U}_1\boldsymbol{V}_1^\top. \tag{22}$$

A third possibility is to map the small singular values to $-1$:

$$\boldsymbol{U}\boldsymbol{V}^\top = \boldsymbol{U}_1\boldsymbol{V}_1^\top - \boldsymbol{U}_2\boldsymbol{V}_2^\top \tag{23}$$

Note that $-\boldsymbol{U}_2\boldsymbol{V}_2^\top$ is in the *opposite* direction as the `Muon` update. If the small singular values carry meaningful information about the loss landscape, then we expect this partly "uphill" step to hurt performance. Comparing the three update rules in Equations (21) to (23) can tell us how small singular values affect `Muon`.

We train GPT-2 Small using each of these three update rules with learning rate 0.05 and weight decay 0.1. We sweep three different options for the cutoff $\gamma$ that defines the 'small' singular values: $10^{-4}$, $10^{-3}$, and $10^{-2}$. The results are plotted in Figure 9. They show that the treatment of singular values smaller than $10^{-4}\sigma_{\max}$ does not matter at all for the performance of `Muon`, and those smaller than $10^{-3}\sigma_{\max}$ have a very minor effect. Notably, even *reversing* the direction of the `Muon` step in the bottom singular subspace barely worsens performance, showing that the gradient information in this subspace not very informative. The bottom panel of Figure 9 shows how five iterations of `Polar Express` (with $\ell = 10^{-3}$) affect small singular values. Singular values greater than $10^{-3}$ are all mapped close to 1, while those smaller than $10^{-4}$ are all mapped close to 0. Thus, while `Polar Express` does not fully converge after five iterations, it does converge in the ways that matter for `Muon`.

**Convergence of Top Singular Values**   As discussed in the previous paragraph, we hypothesize that `Muon` may not be sensitive to the convergence of the small singular values of $\boldsymbol{M}$ when approximating $\mathrm{polar}(\boldsymbol{M})$. Therefore, in Figure 10, we plot the convergence of `Polar Express` and the baselines when all singular values smaller than $10^{-3}$ are ignored. Specifically, if $\mathrm{alg}(\boldsymbol{M})$ denotes the output of an algorithm for approximating $\mathrm{polar}(\boldsymbol{M})$, then we compare

$$\boldsymbol{U}_1\boldsymbol{U}_1^\top \cdot \mathrm{alg}(\boldsymbol{M}) \cdot \boldsymbol{V}_1\boldsymbol{V}_1^\top \qquad \text{to} \qquad \mathrm{polar}_{10^{-3}}(\boldsymbol{M}),$$

where $\mathrm{polar}_{10^{-3}}(\boldsymbol{M}) = \boldsymbol{U}_1\boldsymbol{V}_1^\top = \boldsymbol{U}_1\boldsymbol{U}_1^\top \cdot \mathrm{polar}(\boldsymbol{M}) \cdot \boldsymbol{V}_1\boldsymbol{V}_1^\top$ is the truncated polar factor defined above. The results show that `Polar Express` converges in just six iterations as measured in the relative Frobenius norm and just five iterations when measuring in cosine similarity. The other methods converge faster too, but `Polar Express` still outperforms them. These results may explain why the performance of `Muon` saturates at five or six iterations of `Polar Express`, as shown in Figure 5.

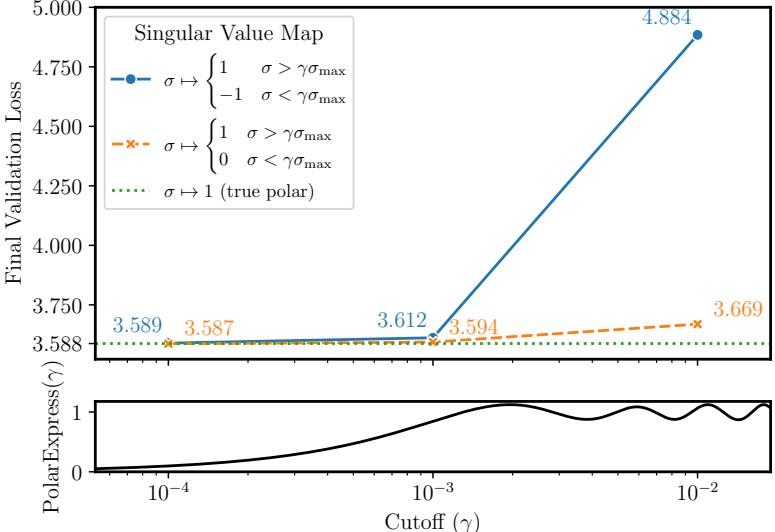

Figure 9: Impact of small singular directions of momentum matrix on optimization quality. We compare three variations of the `Muon` update rule. Exact `Muon` (green) processes the momentum $M = U\Sigma V^\top$ by mapping each singular value to 1: $\mathrm{polar}(M) = UV^\top$. Truncated `Muon` (orange) maps the larger singular values to 1 and the smaller singular values to 0. Reverse `Muon` (blue) maps the larger ones to 1 and the smaller ones to $-1$. Computations are performed in `bfloat32`. All runs train GPT-2 Small on 1 billion tokens of FineWeb data with learning rate 0.05 and weight decay 0.1. When the cutoff that defines "large" and "small" singular values is $\gamma \approx 10^{-3}$, all three methods perform well, showing that the small singular directions do not matter. Bottom panel shows the polynomial defined by composing five iterations of `Polar Express`. Five iterations is just enough for singular values $\geq 10^{-3}$ to nearly converge.

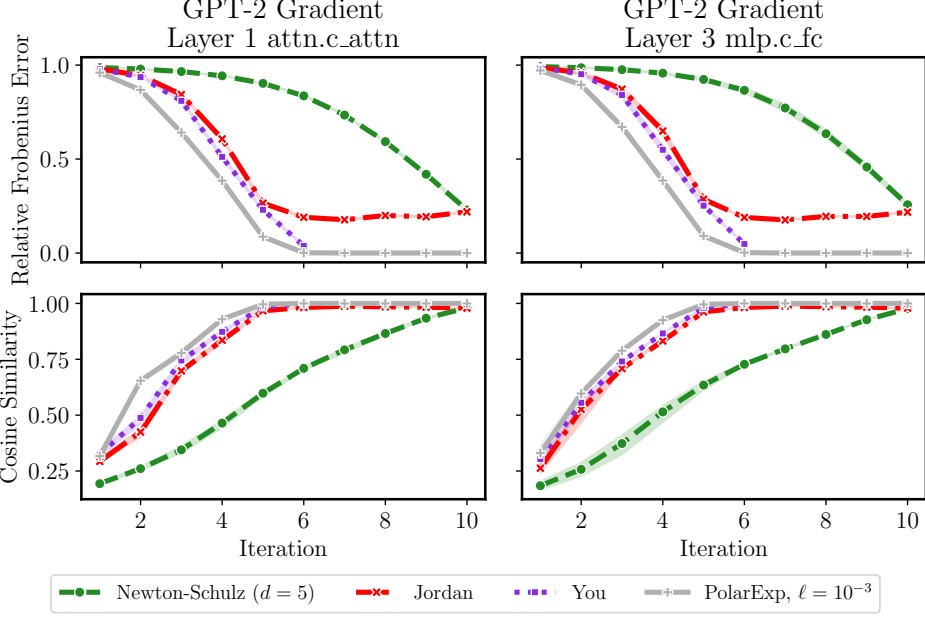

Figure 10: Convergence of degree-5 polynomial methods, considering only singular values larger than $\sigma_{\max}/10^3$. Test matrices are gradients of two layers of a randomly-initialized GPT-2 model on a batch of language modeling data. `Polar Express` converges in just five or six iterations and outperforms other methods.

## H.2 Training GPT-2

**Additional Metrics**  We report additional results from the experiment of Section 4.2. In addition to showing validation loss vs. learning rate and training step, we also report *training* loss vs. learning rate and training *time*. The results are shown in Figures 11a and 11b. The upper rows of each subfigure are identical to Figure 1 and Figure 4, and are repeated here for ease of comparison.

**Weight Decay**  As described in Section 4.3, we reran our GPT-2 training runs with weight decay of 0.1. This change had little effect on the results, as shown in Figure 12.

**Number of Training Tokens**  We also reran some of our GPT-2 training runs using 10 billion tokens of training data instead of 1 billion. As described in Section 4.3, 10 billion tokens roughly matches the Chinchilla scaling rule for GPT-2-Large and exceeds it for GPT-2-Small. Results are shown in Figure 13. Note that the top row of Figure 13a is identical to Figure 6. `Polar Express` still outperforms the baselines across all conditions, but the gap shrinks as the training loss converges.

## H.3 Image Classification

We conducted experiments on the CIFAR-10 and CIFAR-100 image classification benchmarks (Krizhevsky, 2009) using ResNet-20 and ResNet-110 architectures with batch normalization (He et al., 2016). We used a range of learning rates in the range $10^{-6}$ to 1 with a constant learning-rate schedule, a batch size of 128, and 50 epochs of training data. We used three different random seeds for each hyperparameter setting to assess stability and variability. As a baseline, we also included AdamW and SGD with momentum (Kingma & Ba, 2015). Results are given in Figures 14 and 15. For these experiments we see that all the `Muon` variants performed well, matching or exceeding the training loss and validation accuracy of `AdamW` and `sgd-m` while also being more stable with respect to the choice of learning rate. However, we do not see a marked difference between the varieties of `Muon`. Indeed, even Newton-Schulz (degree = 5) performs equally well in this context, despite being significantly less accurate than `PolarExpress`, `Jordan` or `You`.

Next we train a Vision Transformer (patch size 4, embedding dimension 512, depth 6, 8 heads, MLP dimension 512, dropout 0.1) on CIFAR-10 for 200 epochs with batch size 512 using a constant learning rate schedule. Results are shown in Figure 16. `Muon` with `Polar Express` achieved the best training and validation loss (closely followed by Jordan's and You's methods). However, improved loss did not entirely translate to better accuracy: both `Muon` and Newton-Schulz and Adam performed well in terms of validation accuracy. Overall, these experiments do not show a consistent advantage for `Polar Express`. Further work may be beneficial to fully realize the potential benefits of `Muon` and to further tune `Polar Express` for these settings.

## I  Initialization for Matrices with Large Spectral Gaps

In Section 3, we constructed a sequence of polynomials that is adapted to the range of the singular values $[\ell, u]$. Assuming nothing else about the input, these polynomials are optimal since they provide a good approximation to 1 across the entire interval. However, in many applications, the spectrum has large gaps; that is, there are several large outlying singular values that are well-separated from the rest. For these matrices, it is not necessary for the polynomial to be accurate on the entire interval $[\ell, u]$, only on the range of the small singular values plus a few other isolated points. In this section, we take advantage of this structure to accelerate our method by preprocessing the matrix to eliminate the largest singular values.

The first step is to find small intervals containing each of these large singular values. To find lower bounds, we use subspace iteration, which is a generalization of the power method that approximates multiple singular values simultaneously. Fix $k$, the number of singular values we wish to eliminate. Letting $\sigma_1 \geq \cdots \geq \sigma_n$ denote the singular values of $M$, subspace iteration produces estimates

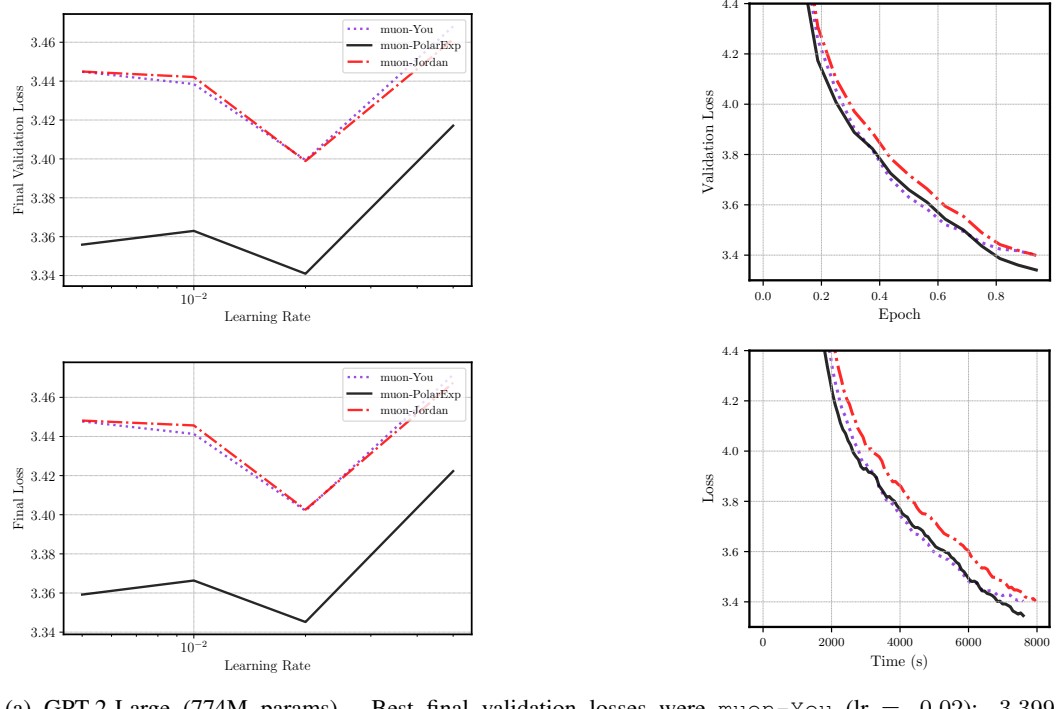

(a) GPT-2-Large (774M params). Best final validation losses were `muon-You` (lr = 0.02): 3.399, `muon-Jordan` (lr = 0.02): 3.398 and `muon-PolarExp` (lr = 0.02): 3.340.

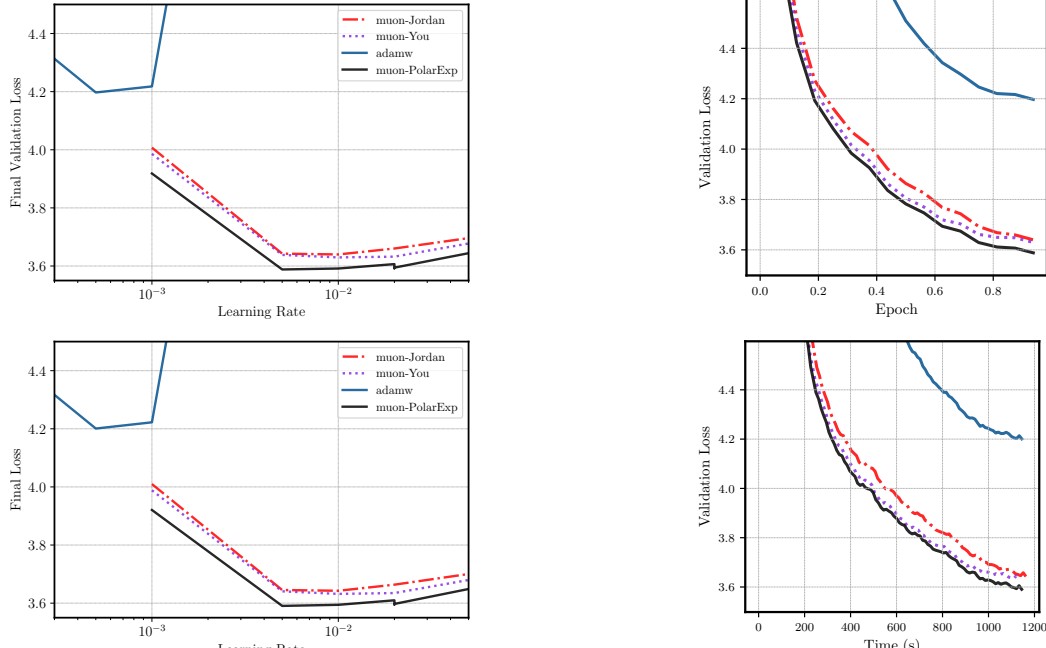

(b) GPT-2-Small (124M params). Best final validation losses were `adamw` (lr = 0.001): 4.197, `muon-Jordan` (lr = 0.01): 3.639, `muon-You` (lr = 0.01): 3.629 and `muon-PolarExp` (lr = 0.005): 3.588.

Figure 11: Training GPT-2 on 1 billion tokens of FineWeb data (Penedo et al., 2024) without weight decay. The label muon-<method> denotes `Muon` with 5 iterations of <method> to compute polar($M$). Top left: final validation loss vs. learning rate. Bottom left: final training loss vs. learning rate. Top right: validation loss vs. number of iterations for best learning rate. Bottom right: training loss vs. time for best learning rate.

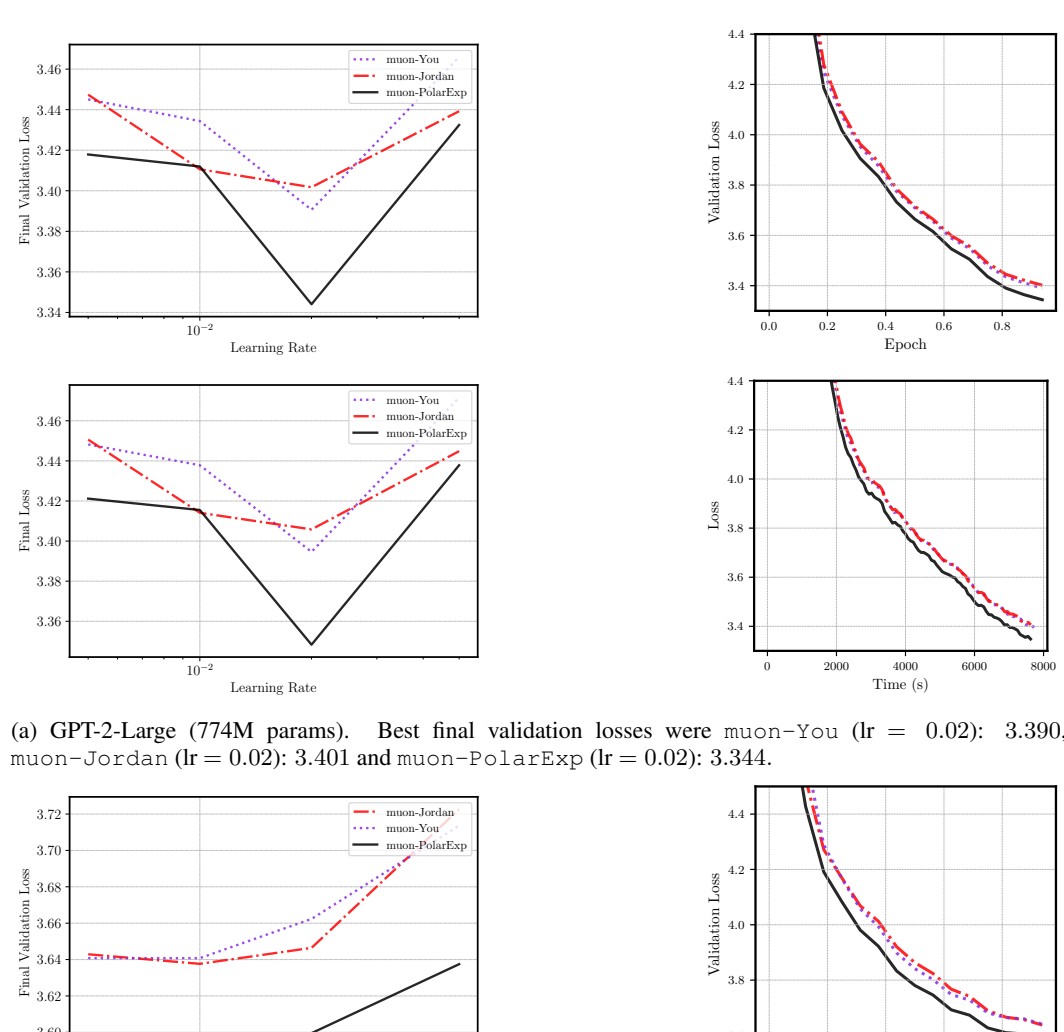

(a) GPT-2-Large (774M params). Best final validation losses were `muon-You` (lr = 0.02): 3.390, `muon-Jordan` (lr = 0.02): 3.401 and `muon-PolarExp` (lr = 0.02): 3.344.

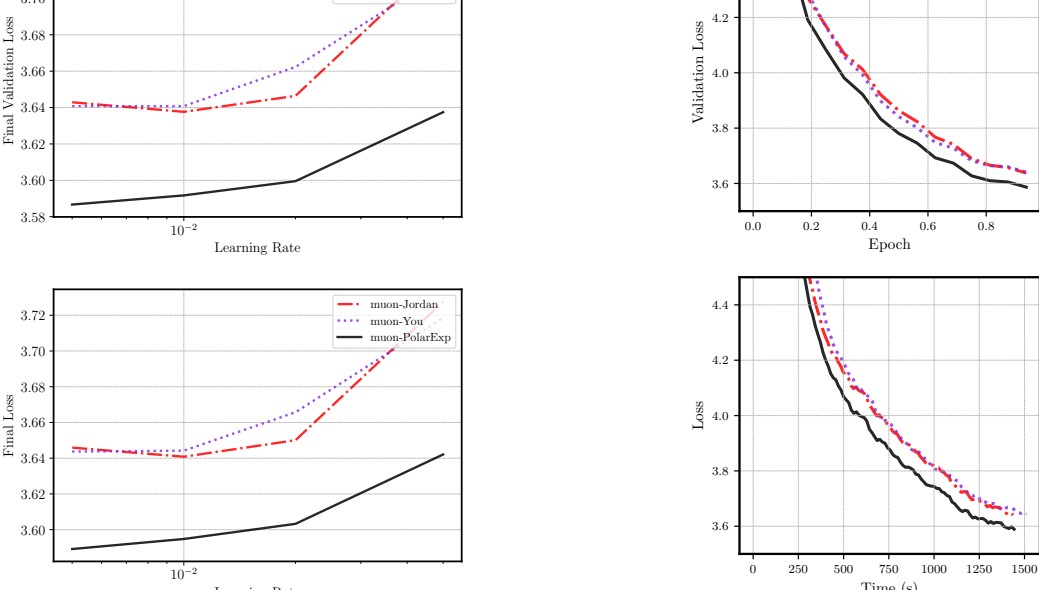

(b) GPT-2-Small (124M params). Best final validation losses were `muon-Jordan` (lr = 0.01): 3.638, `muon-You` (lr = 0.005): 3.641 and `muon-PolarExp` (lr = 0.005): 3.587.

Figure 12: Training GPT-2 on 1 billion tokens of FineWeb data (Penedo et al., 2024) **with weight decay** 0.1. The label muon-<method> denotes Muon with 5 iterations of <method> to compute polar($M$). Top left: final validation loss vs. learning rate. Bottom left: final training loss vs. learning rate. Top right: validation loss vs. number of iterations for best learning rate. Bottom right: training loss vs. time for best learning rate.

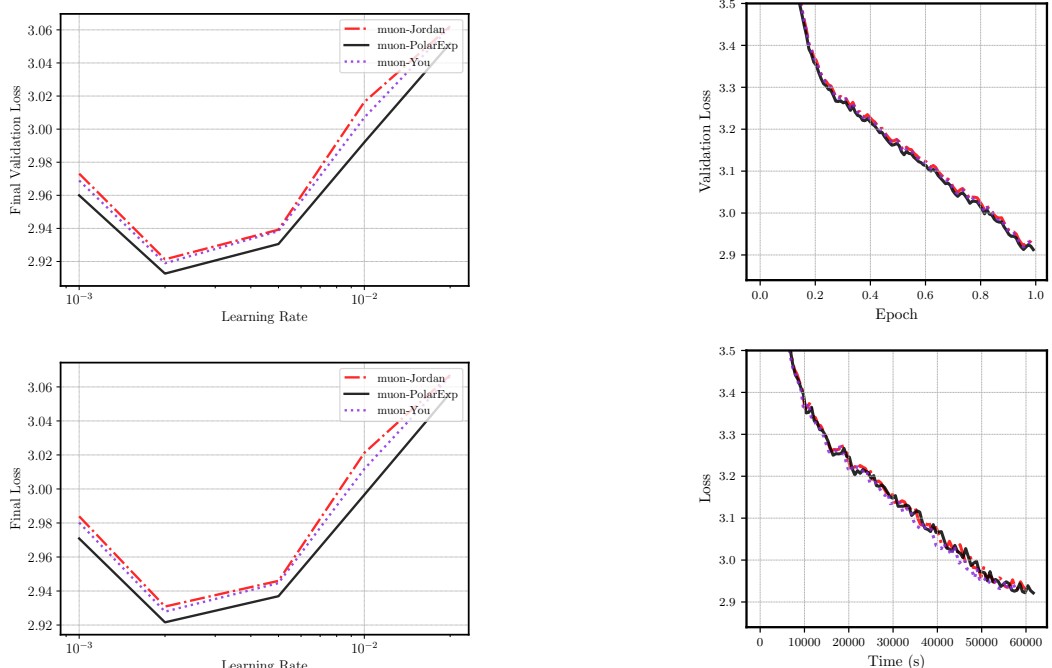

(a) GPT-2-Large (774M params) with weight decay 0.1. Best final validation losses were `muon-Jordan` (lr = 0.002): 2.921, `muon-You` (lr = 0.002): 2.919 and `muon-PolarExp` (lr = 0.002): 2.913.

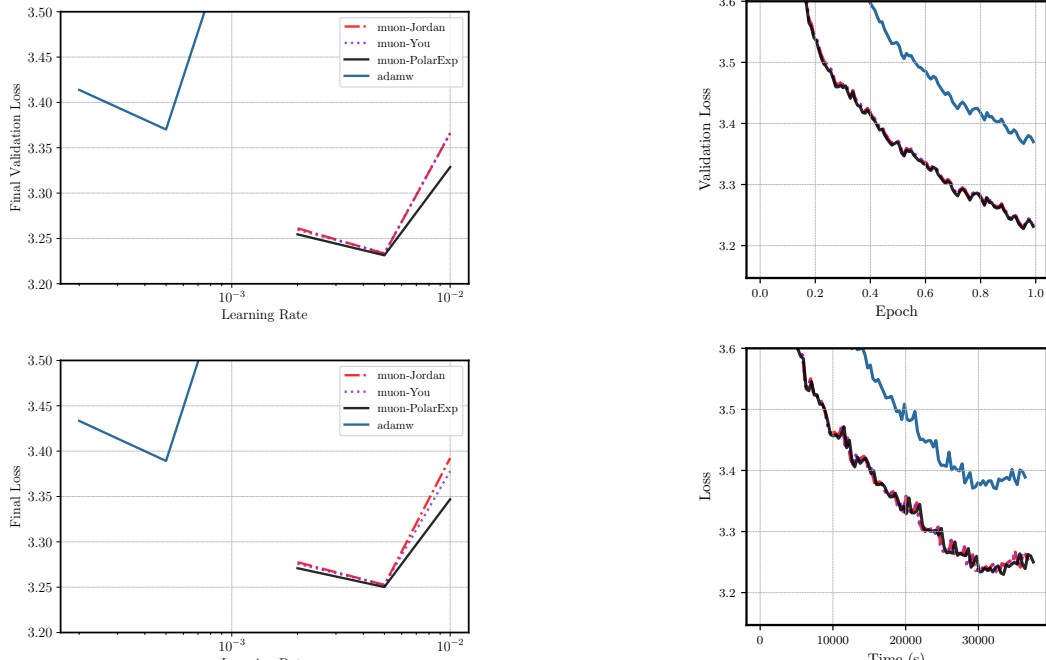

(b) GPT-2-Small (124M params) without weight decay. Best final validation losses were `adamw` (lr = 0.0005): 3.370, `muon-Jordan` (lr = 0.005): 3.233, `muon-You` (lr = 0.005): 3.234 and `muon-PolarExp` (lr = 0.005): 3.231.

Figure 13: Training GPT-2 **on 10 billion tokens of FineWeb data** (Penedo et al., 2024). The label muon-<method> denotes Muon with 5 iterations of <method> to compute $\text{polar}(M)$. Top left: final validation loss vs. learning rate. Bottom left: final training loss vs. learning rate. Top right: validation loss vs. number of iterations for best learning rate. Bottom right: training loss vs. time for best learning rate.

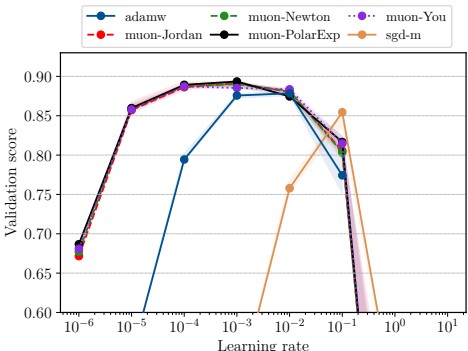
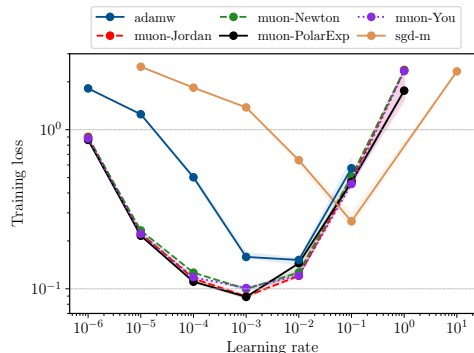

Figure 14: `CIFAR10` with a `RESNET20`. Shaded regions show range over three random seeds. The best validation accuracy for each method was `sgd-m` (lr = 0.1): 0.855 `Adamw` (lr = 0.01): 0.878 `muon-You` (lr = 0.001): 0.887, `muon-Newton` (lr = 0.001): 0.890, `muon-Jordan` (lr = 0.001): 0.891, `muon-PolarExp` (lr = 0.001): 0.893.

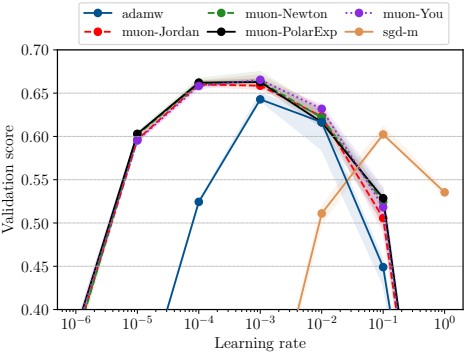
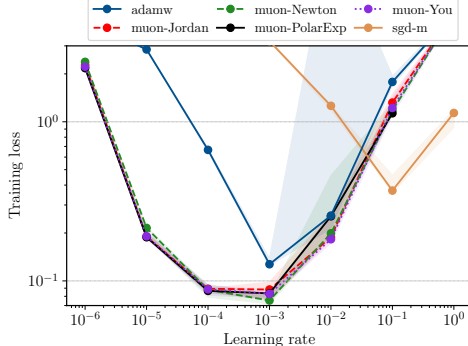

Figure 15: `CIFAR100` with `RESNET110`. Shaded regions show range over three random seeds. The best validation accuracy for each method was `sgd-m` (lr = 0.1): 0.602, `Adamw` (lr = 0.01): 0.643, `muon-Jordan` (lr = 0.001): 0.660, `muon-Newton` (lr = 0.001): 0.663. `muon-PolarExp` (lr = 0.001): 0.663, `muon-You` (lr = 0.001): 0.665,

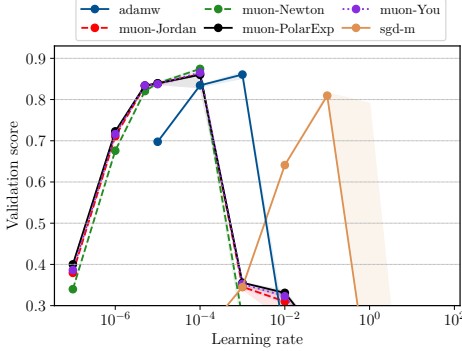
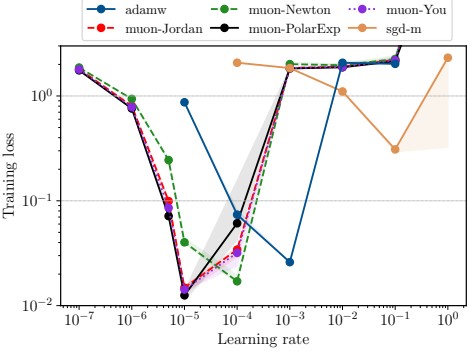

Figure 16: `CIFAR10` with a `VIT`. Shaded regions show range over three random seeds. The best validation accuracy for each method was `sgd-m` (lr = $10^{-1}$): 0.809, `muon-PolarExp` (lr = $10^{-5}$): 0.860, `Adamw` (lr = $10^{-3}$): 0.861, `muon-Jordan` (lr = $10^{-5}$): 0.861, `muon-You` (lr = $10^{-5}$): 0.865, `muon-Newton` (lr = $10^{-4}$): 0.874 .

$\tilde{\sigma}_1 \geq \cdots \geq \tilde{\sigma}_k$ satisfying $\sigma_i \geq \tilde{\sigma}_i$ for all $i \in 1, \ldots, k$.[10] To find upper bounds on each $\sigma_i$, we can use the fact that $\|\boldsymbol{M}\|_{\mathrm{F}}^2 = \sum_{j=1}^n \sigma_j^2$ as follows:

$$\sigma_i^2 = \|\boldsymbol{M}\|_{\mathrm{F}}^2 - \sum_{\substack{j=1 \\ j \neq i}}^n \sigma_j^2 \leq \|\boldsymbol{M}\|_{\mathrm{F}}^2 - \sum_{\substack{j=1 \\ j \neq i}}^k \sigma_j^2 \leq \|\boldsymbol{M}\|_{\mathrm{F}}^2 - \sum_{\substack{j=1 \\ j \neq i}}^k \tilde{\sigma}_j^2 \tag{24}$$

That is, for each $i \in [n]$,

$$\sigma_i \in \left[ \tilde{\sigma}_i, \sqrt{\|\boldsymbol{M}\|_{\mathrm{F}}^2 - \sum_{\substack{j=1 \\ j \neq i}}^k \tilde{\sigma}_j^2} \right]$$

Setting $i = k+1$, the above also provides an upper bound for the tail of the spectrum, $\sigma_{k+1}, \ldots, \sigma_n$.

The second step is to find an odd polynomial that well-approximates the constant function on each of these intervals and on the tail simultaneously. For simplicity, we treat only the $k = 1$ case here. Assume that $\boldsymbol{M}$ is normalized to $\|\boldsymbol{M}\|_{\mathrm{F}} = 1$ and let $z = \tilde{\sigma}_1$ be the lower bound produced by subspace iteration (which reduces to the power method in this case). Then (24) gives $\sigma_1 \in [z, 1]$ and $\sigma_2, \ldots, \sigma_n \leq \sqrt{1 - z^2}$. Assume that these intervals do not overlap, that is, $\sqrt{1 - z^2} \leq z \iff z \geq 1/\sqrt{2}$. Then we construct the unique odd cubic polynomial $p(x) = ax + bx^3$ that satisfies $p(\sqrt{1 - z^2}) = 1$ and $p(z) = 1$ by setting

$$a = \frac{z^2(z + \sqrt{1 - z^2}) - \sqrt{1 - z^2}}{z\sqrt{1 - z^2}(2z^2 - 1)} \qquad b = \frac{\sqrt{1 - z^2} - z}{z\sqrt{1 - z^2}(2z^2 - 1)} \tag{25}$$

Because $p(0) = 0$ and $p$ has at most one local extremum on $\mathbb{R}_{\geq 0}$, these conditions immediately guarantee that $p$ is concave-increasing on $[0, \sqrt{1 - z^2}]$, so it must lie above the line $x \mapsto x/\sqrt{1 - z^2}$. Furthermore, $p$ is decreasing on $[\sigma_1, 1]$, so it maps $\sigma_1 \in [z, 1]$ to $[p(1), 1]$. By minimizing $p(1)$ over all valid $z$ (that is, over the interval $z \in [1/\sqrt{2}, 1]$), one can further show that $p(1) > 1/\sqrt{2}$, so $\sigma_1$ cannot be decreased very much by applying $p$. Thus, the largest singular value of $p(\boldsymbol{M})$ is still at most 1, while the smaller singular values have increased by a potentially large factor of $1/\sqrt{1 - z^2}$. When there is a large outlying singular value, $z$ is close to 1 and this initialization scheme makes much more progress than a standard iteration of `PolarExpress` would have.

In Figure 17, we demonstrate the benefit of using the $p$ given by (25) on a synthetic matrix whose spectrum follows a power law decay. That is, $\sigma_j(\boldsymbol{M}) = j^{-5}$, so this matrix has a large outlying singular value $\sigma_1 \gg \sigma_2$. Applying (25) costs almost as much as performing an iteration of a degree-5 polynomial method, so for fair comparison, we count it as an additional iteration in this plot. For both Newton-Schulz and `Polar Express`, performing the extra spectrum-aware initialization step described in this section leads to significant speedups in convergence.

## J  FAST POLYNOMIAL ITERATION FOR RECTANGULAR MATRICES

In this section, we describe a simple method for applying an iterative polynomial method to a rectangular matrix. For matrices with a large aspect ratio, this method yields significant computational savings. We emphasize that this method is applicable to *any* computation of the form $(p_T \circ \cdots \circ p_1)(\boldsymbol{X})$, where each $p_t$ is an odd polynomial. Thus, it can be used to apply Newton-Schulz or Jordan's polynomials in addition to our own.

As a preliminary, we first describe the baseline approach. Let $\boldsymbol{X} \in \mathbb{R}^{m \times n}$ with $m \geq n$, where $\alpha := m/n \geq 1$ is called the aspect ratio. Any odd polynomial $p$ of degree $d = 2q + 1$ can be represented as $p(x) = xh(x^2)$, where $h$ is a polynomial of degree $q$. Thus, $p(\boldsymbol{X}) = \boldsymbol{X}h(\boldsymbol{X}^\top \boldsymbol{X})$. Furthermore, $h$ can be written in a factored form called Horner's rule to reduce the number of multiplications. For instance, if $h(y) = a + by + cy^2 + dy^3$, Horner's rule gives $h(y) = a + y(b + y(c + dy))$. For a matrix, $h(\boldsymbol{Y}) = a\boldsymbol{I} + \boldsymbol{Y}(b\boldsymbol{I} + \boldsymbol{Y}(c\boldsymbol{I} + d\boldsymbol{Y}))$. Thus for $\boldsymbol{Y} \in \mathbb{R}^{n \times n}$, computing $h(\boldsymbol{Y})$ costs about

---

[10]Let $\boldsymbol{Q}_0 \in \mathbb{R}^{n \times k}$ be a random matrix with orthonormal columns and define $\boldsymbol{Q}_{t+1}, \boldsymbol{R}_{t+1} = \mathrm{qr}(\boldsymbol{M}^\top \boldsymbol{M} \boldsymbol{Q}_t)$, where qr is the QR decomposition. Subspace iteration outputs the singular values $\tilde{\sigma}_1, \ldots, \tilde{\sigma}_k$ of $\boldsymbol{M}\boldsymbol{Q}_T, \tilde{\sigma}_1, \ldots, \tilde{\sigma}_k$. By the Cauchy interlacing theorem, $\tilde{\sigma}_k \leq \sigma_k$.

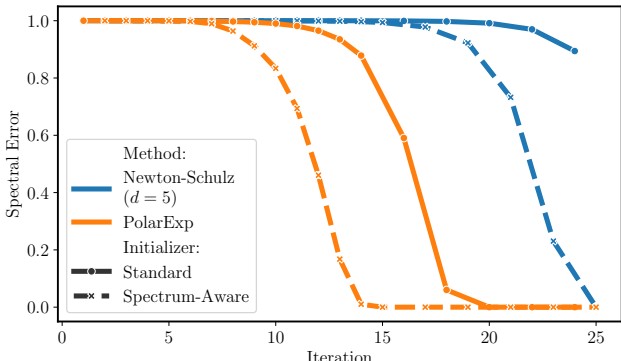

Figure 17: Benefits of the spectrum-aware initialization scheme of Appendix I. Using this scheme improves convergence of both Newton-Schulz and `Polar Express` on a synthetic $32 \times 32$ matrix with $\sigma_j(\boldsymbol{M}) = j^{-5}$. Note that we count the spectrum-aware initialization as an additional iteration.

$(\deg(h) - 1) \cdot n^3$ operations, and computing $p(\boldsymbol{X}) = \boldsymbol{X} h(\boldsymbol{X}^\top \boldsymbol{X})$ costs $2mn^2 + \left(\frac{d-1}{2} - 1\right) \cdot n^3 = \left(\frac{d-3}{2} + 2\alpha\right) \cdot n^3$ operations. This process could be repeated for each iteration $p_1, \ldots, p_T$. Notice that if we instead computed $h(\boldsymbol{X} \boldsymbol{X}^\top) \boldsymbol{X}$, the result would be the same but the cost would be higher.

A major drawback of this naive approach is that it has a strong dependence on $\alpha$, since two rectangular matrix multiplications must be performed in *each* of the $T$ iterations. When $m \gg n$, these two multiplications dominate the cost. In Algorithm 3, we introduce a simple trick that dramatically reduces this cost, using just two rectangular matrix multiplications to compute *all* $T$ iterations.

---

**Algorithm 3** Fast Polynomial Iteration for Rectangular Matrices

---

**input:** $\boldsymbol{X} \in \mathbb{R}^{m \times n}$ with $m > 1.5n$, odd polynomials $p_1(x) = x h_1(x^2), \ldots, p_T(x) = x h_T(x^2)$.
**output:** The matrix $(p_T \circ \cdots \circ p_1)(\boldsymbol{X})$.
   $\boldsymbol{Y} = \boldsymbol{X}^\top \boldsymbol{X}$                                                        $\triangleright mn^2$
   Let $\boldsymbol{Q}_0 = \boldsymbol{I}$
   **for** $t = 1, 2, \ldots, T$ **do**
      $\boldsymbol{R}_t = \boldsymbol{Q}_{t-1}^\top \boldsymbol{Y} \boldsymbol{Q}_{t-1}$                                             $\triangleright 2n^3$
      $\boldsymbol{Q}_t = \boldsymbol{Q}_{t-1} h_t(\boldsymbol{R}_t)$                        $\triangleright$ Horner's rule: $\deg(h_t) \cdot n^3$
   **end for**
   **return** $\boldsymbol{X} \boldsymbol{Q}_T$                                              $\triangleright mn^2$

---

To see why this works, define $q_0(x) = 1$,

$$q_t(x) = \frac{(p_t \circ \cdots \circ p_1)(x)}{x} = \frac{p_t\left((p_{t-1} \circ \cdots \circ p_1)(x)\right)}{x} = \frac{p_t\left(x q_{t-1}(x)\right)}{x} \tag{26}$$

$$= \frac{x q_{t-1}(x) \cdot h_t\left((x q_{t-1}(x))^2\right)}{x} = q_{t-1}(x) \cdot h_t\left(x^2 \cdot q_{t-1}(x)^2\right) \tag{27}$$

and $r_t(x) = x^2 \cdot q_{t-1}(x)^2$. It is clear by induction that $\boldsymbol{R}_t = r_t(\boldsymbol{X}), \boldsymbol{Q}_t = q_t(\boldsymbol{X})$, and $\boldsymbol{X} \boldsymbol{Q}_T = (p_t \circ \cdots \circ p_1)(\boldsymbol{X})$. As promised, this algorithm uses no rectangular multiplications in the for-loop. If each $p_t$ is degree $d$, then the total cost is $\left(\frac{d+3}{2}T + 2\alpha\right) \cdot n^3$. When $\alpha > 1.5\frac{T}{T-1}$, this is smaller than the naive method. We can use this criterion to select either Algorithm 3 or the baseline method at runtime.[11]

Algorithm 3 can introduce numerical errors, especially when working in a low precision format like `bfloat16`. We identify two sources of numerical trouble and propose remedies for each. The first is due to the ill-conditioning of $\boldsymbol{X}$. Let $\boldsymbol{X} = \boldsymbol{U} \boldsymbol{\Sigma} \boldsymbol{V}^\top$ be the SVD. For large $T$, $(p_T \circ \cdots p_1)(\boldsymbol{X}) = \boldsymbol{X} \boldsymbol{Q}_T \approx \text{polar}(\boldsymbol{X}) = \boldsymbol{U} \boldsymbol{V}^\top$. Thus, $\boldsymbol{Q}_T \approx \boldsymbol{V}^\top \boldsymbol{\Sigma}^{-1} \boldsymbol{V}$. When $\boldsymbol{X}$ has very small

---

[11]Notice that $\boldsymbol{Q}_T \to \boldsymbol{Y}^{-1/2}$. This shows that the `Polar Express` polynomials also give a method of computing the inverse square root of a PSD matrix.

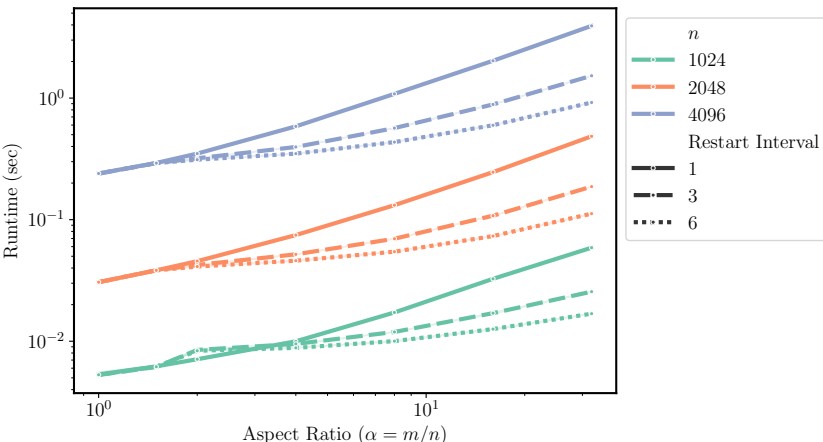

Figure 18: Effects of using Algorithm 3 on runtime on a GPU. We run $T = 6$ iterations of a degree-5 polynomial method on matrices with various dimensions $n$ and aspect ratios $\alpha$. Restart interval $= 6$ is Algorithm 3, restart interval $= 1$ is equivalent to the baseline (that is, not using Algorithm 3), and restart interval $= 3$ is an intermediate method that calls Algorithm 3 once to do the first three iterations and again to do the last three iterations for greater stability. When $\alpha \gg 1$, increasing the restart interval *significantly* reduces the runtime.

singular values and the floating point precision is very low, instantiating $\boldsymbol{Q}_T$ may be unstable. To mitigate this issue, we use a restarting strategy. Notice that the issue arises only for large $T$, for which $(p_T \circ \cdots \circ p_1)(\epsilon) \approx 1$. Limiting ourselves to $T = 3$ iterations improves the conditioning of $\boldsymbol{Q}_T$ because $(p_T \circ \cdots \circ p_1)(\epsilon) \ll 1$. Thus, to compute $T > 3$ iterations, we begin with $\boldsymbol{X}_0$ and apply Algorithm 3 with the first three polynomials, producing $\boldsymbol{X}_3$. When then apply Algorithm 3 again with the next three polynomials to $\boldsymbol{X}_3$, producing $\boldsymbol{X}_6$, and so on. As $\boldsymbol{X}_t$ approaches convergence, its conditioning improves and we may no longer need to restart at all. Note that restarting Algorithm 3 after every iteration is exactly the same as the baseline method.

Second, while the matrix $\boldsymbol{Y}$ is positive definite in exact arithmetic, numerical round-off can introduce spurious negative eigenvalues that cause the method to diverge to infinity. To combat this issue, we instead set $\boldsymbol{Y} = \boldsymbol{X}^\top \boldsymbol{X} + 10^{-3}\boldsymbol{I}$ during the first application of Algorithm 3. (We also normalize by $\|\boldsymbol{X}\|_{\mathrm{F}} + 10^{-3}$ instead of $\|\boldsymbol{X}\|_{\mathrm{F}}$.) In subsequent restarts of Algorithm 3, we set $\boldsymbol{Y} = \boldsymbol{X}^\top \boldsymbol{X}$ as before. This is akin to slightly increasing each of the singular values of $\boldsymbol{X}$, but it does *not* change the polar factor of $\boldsymbol{X}$. Thus, while the output will be slightly different in the early iterations, the algorithm still converges to the correct answer.

Figure 18 shows that using Algorithm 3 can significantly improve runtime on the GPU when the aspect ratio is large enough. As expected, using Algorithm 3 for many iterations significantly reduces the dependence of the runtime on the aspect ratio. Running six iterations of a degree-5 polynomial method when $\alpha = 4$ (as with the linear transformations in each MLP block of a transformer) we obtain almost a 2x speedup, and when $\alpha = 32$, we obtain a 5x speedup. If we restart every three iterations, the trend is the same but the runtime savings are somewhat smaller.

### J.1 APPLICATION TO MUON

If these problems can be mitigated, the speed afforded by Algorithm 3 suggests an improvement in the way Muon is applied to transformers. In sum, the idea is to replace one large matrix with a small aspect ratio by many smaller matrices with large aspect ratios and apply Algorithm 3 to all of them in parallel. Each multi-head attention layer contains four square weight matrices $\boldsymbol{W}_Q, \boldsymbol{W}_K, \boldsymbol{W}_V$ and $\boldsymbol{W}_O \in \mathbb{R}^{d \times d}$. The orthogonalization step of Muon is either applied separately to these four matrices or else to $[\boldsymbol{W}_Q \mid \boldsymbol{W}_K \mid \boldsymbol{W}_V]$ and $\boldsymbol{W}_O$, since typical implementations of multi-head attention store the weights in this concatenated form. However, we believe it is natural to consider each of these four weight matrices to be a concatenation of many smaller linear transformations, each corresponding to a single attention head. If $H$ is the number of heads, each of these smaller matrices has size

$d \times \frac{d}{H}$; that is, they have aspect ratio $\alpha = H$. The gradient matrices of $[\boldsymbol{W}_Q \mid \boldsymbol{W}_K \mid \boldsymbol{W}_V]$ and $\boldsymbol{W}_O$ can be reshaped into 3-tensors in which each slice is one of these smaller matrices. Since typical transformers like GPT-3 can have as many as 96 heads, this variation of `Muon` has the potential to reduce the runtime.

We use this idea to train a GPT-Small model on FineWeb1B. We compare four conditions:

1. The baseline approach used in the rest of this paper (not splitting $[\boldsymbol{W}_Q \mid \boldsymbol{W}_K \mid \boldsymbol{W}_V]$ and not using Algorithm 3)
2. Splitting up the gradient matrices of $[\boldsymbol{W}_Q \mid \boldsymbol{W}_K \mid \boldsymbol{W}_V]$ and $\boldsymbol{W}_O$ by head and applying Muon to each piece, as described above
3. Using Algorithm 3, restarted after three iterations, on all rectangular weight matrices
4. Splitting by head *and* using Algorithm 3

We used `Polar Express` with weight decay of 0.1 for all conditions and swept learning rates $0.003, 0.005, 0.01$. Otherwise, all hyperparameters were the same as in Section 4.2.

Our results showed that these changes had a negligible effect in this setting. They did not affect the optimization quality. Compared to the baseline, splitting by heads actually reduced the final loss slightly from 3.59 to 3.55; using Algorithm 3 increased the loss very slightly, from 3.59 to 3.60 when not splitting by head, and from 3.55 to 3.56 when we did split. However, the runtimes of all 12 runs were nearly identical, showing that at this scale, the FLOP savings of Algorithm 3 is not beneficial. The embedding size of GPT-Small is just 768. These techniques may be more impactful when using a larger model. It may also have more impact outside of deep learning, where `Polar Express` would be run for more than the 5 iterations used in our experiments. We leave exploration of these settings to future work.

