# OpenReview forum: "The Polar Express: Optimal Matrix Sign Methods and their Application to the Muon Algorithm"
_ICLR.cc/2026/Conference — ICLR 2026 Oral_

### Official Review · Reviewer_WacA · 2025-10-29

**Soundness:** 3
**Presentation:** 3
**Contribution:** 3
**Rating:** 6
**Confidence:** 3

**Summary:**

This paper proposes a new method for orthogonalizing a matrix, with applications in the Muon optimizer. The method, PolarExpress, greedily computes polynomial approximations of the sign function in a residual manner using the Remez algorithm, and then applies these approximations during training to approximately orthogonalize the momentum buffer in Muon. The authors prove results about the convergence of PolarExpress and its optimality with respect to the L_\infty norm to the sign function, and empirically show that PolarExpress improves upon existing orthogonalization methods used in current Muon implementations.

**Strengths:**

I will preface this review by saying that I am not an expert in numerical methods. However, the presented algorithm seems novel and interesting, and the there is theory showing that PolarExpress is guaranteed to converge at least as fast as the canonical "NewtonShultz5"  method. This paper also has some other nice components, including
- Theoretical justification for using a greedy residual polynomial construction
- Empirical results on toy matrices showing that the method converges faster than existing baselines (Figure 3)
- Results showing the effect of varying the lower bound guess $\ell$ of the smallest singular value in the gradient matrix
- Empirical GPT training experiments showing that PolarExpress achieves better initial convergence over prior Muon implementations.

**Weaknesses:**

- There seems to be a missing "key" baseline, which is running Muon with the actual `polar' function and computing the SVD online. Although this is obviously impractical, it gives a sense of how far PolarExpress and these other methods are from "true" Muon. Can you run an experiment showing how much benefit there is from getting the optimal solution in real-world GPT training?
- Likewise, does minimizing the L_\infty norm of the sign function maximize the directional similarity to the actual "true" Muon update? Can you generate a plot of the cosine distance of PolarExpress's update (and other baselines) to the SVD update to get a sense of how close these approximations are?
- The empirical experiments only go out to 1B tokens. 125M parameters @ 1B tokens is way below Chinchilla optimal, and my own experiments in the past have suggested that Muon's initial performance gap diminishes in the "overtrained regime." If you have time, can you run an experiment past Chinchilla (125M parameters @ 10B tokens should be easily doable in a day on the 4 H100s used in the paper) to see how much of a gap there is?

**Questions:**

See above.

---

> ### Author Response · Authors · 2025-11-21
>
> Thank you so much for the thoughtful feedback! We think your questions are very astute, and we adopted each of your suggestions to improve our paper. We believe that the new experiments and figures in our latest version give a clear answer to each of them. We will address them in turn.
>
> ## Question 1: How far is PolarExpress from the “true” Muon?
>
> We took your suggestion and added a new experiment to Section 4.3. See Figure 5. We train a GPT-2-Small model using the actual polar function, as computed via an SVD. We also run Polar Express with a varying number of iterations, from 2 to 30. Recall that our main experiments use 5 iterations. Our findings show that increasing the accuracy of computing $\mathrm{polar}(\mathbf M)$ is not worthwhile in this setting. Beyond 5 or 6 iterations of Polar Express, the loss of the model trained with Muon does not improve. This is true of the SVD too. Our experiments also confirm that the SVD is significantly slower than Polar Express, dominating even the cost of computing the forward and backward passes.
>
>
> ## Question 2: Does minimizing the L_\infty norm of the sign function maximize the directional similarity to the actual "true" Muon update? Can you generate a plot of the cosine distance of PolarExpress's update?
>
> We have added an additional plot in Appendix H (Figure 8) showing the cosine similarity and Frobenius norm error of our approximation. This plot supplements the spectral norm error shown in Figure 3. See the heading “Convergence in Frobenius Norm” in Appendix H. The results are similar in all three metrics. Note that cosine similarity and Frobenius norm are conceptually similar, since they are both defined in terms of the Frobenius inner product. However, while cosine similarity measures the angle between two matrices but ignores differences in magnitude, the Frobenius norm measures both direction and magnitude.
>
> We next address the relationship between the cosine similarity and the $L_{\infty}$ error of our approximation to $\mathrm{sign}(x)$ — or equivalently, the worst-case spectral norm error of our approximation to $\mathrm{polar}(\mathbf M)$. These two metrics are not equivalent, but they are closely related: the spectral norm error $\|\|p(M) - \mathrm{polar}(M)\|\|_2$ measures how far the singular values of $p(M)$ are from 1. The cosine similarity measures how far the singular values of $p(M)$ are from one another.
>
> We now derive an expression for the cosine similarity in terms of the singular values of $M$. Let $M = U \Sigma V^T$ with $\mathrm{rank}(M) = \mathrm{rank}(\Sigma) = r$. Consider a polynomial approximation $X = U p(\Sigma) V^T$ to $\mathrm{polar}(M) = U V^T$. The cosine distance between $X$ and $\mathrm{polar}(M)$ is $\cos(\theta) = \frac{\langle \mathrm{polar}(M), X \rangle}{\|\|\mathrm{polar}(M)\|\|_F \|\|X\|\|_F}$. Since $\|\|\mathrm{polar}(M)\|\|_F = \sqrt{\mathrm{rank}(M)} = \sqrt{r}$ and $M$ and $X$ share singular vectors we obtain $\cos(\theta) = \frac{\mathrm{tr}(p(\Sigma))}{\sqrt{r}\|\|p(\Sigma)\|\|_F} = \frac{\sum_i p(\sigma_i)}{\sqrt{r\sum_i p(\sigma_i)^2}}$. Thus, $\cos(\theta) = 1$ when $p(\sigma_1) = \cdots = p(\sigma_r) > 0$. Thus, cosine similarity measures how far the singular values of $p(M)$ are from one another. For comparison, $\|\|p(M) - \mathrm{polar}(M)\|\|_2 = \max_i |p(\sigma_i) - 1|$.
>
> We  now provide a justification for measuring the error of our approximation using the spectral norm/$L_{\infty}$ norm. The action of a single layer with weight matrix $W$ is $x \mapsto Wx$. Let $W_{t+1} = W_{t} - \lambda \mathrm{polar}(M_t)$ be the “true” Muon update of the weights at iteration $t+1$, where $\lambda$ is the learning rate and $M_t$ is the momentum matrix, and let $\hat W_{t+1} = W_{t} - \lambda p(M_t)$ denote the approximate Muon update. Then, for any input $x$, $\|\|\hat W_{t+1} x - W_{t+1}x\|\|_2 = \lambda \|\|p(M_t) x- \mathrm{polar}(M_t)x\|\|_2 \leq \lambda \|\|p(M_t) - \mathrm{polar}(M_t)\|\|_2\|\|x\|\|_2$. Hence, minimizing the $L$$\infty$ norm error directly minimizes the worst-case difference in the action of the approximate Muon update compared to the true one.
>
> ## Question 3. Can you run an experiment on a model past Chinchilla to see how much of a gap there is?
>
> We ran the experiment you suggested, training both GPT-Large-2 and GPT-Small-2 on 10 billion tokens instead of 1 billion. For GPT-Large-2, this is roughly Chinchilla optimal, while for GPT-Small-2 it is squarely in the overtrained regime. The results appear in Figure 11 of Appendix H, and a subset of the results appears in Figure 6 in the main text. The results show a consistent advantage of Polar Express over the baseline methods across all learning rates. However, as you suggest, the gap between the different methods is smaller in the overtrained regime. In the overtrained regime, as the model gets closer to its irreducible loss, it may be more difficult to distinguish small differences in the performance of optimization methods.

---

> > ### Comment · Reviewer_WacA · 2025-11-21
> >
> > This is very interesting. I'm surprised that the cosine similarity empirically saturates *after* 5 iterations, but that 5 iterations seems to be sufficient to get very close to SVD. Did you save the checkpoints for the various iteration runs and the SVD run? If so and if they're the same seed, can you plot the KL between model outputs over some devset for different pairs of runs? It would be interesting to know if varying the number of iterations changes the final model significantly or all of these runs are converging to similar points.

---

> > > ### Author Response · Authors · 2025-12-03
> > >
> > > The reviewer raises an interesting question. From looking at the convergence plots (Figures 3 and 8 in the latest revision) it is clear that none of the polar methods converges until at least 7 or 8 iterations. However, our ablation experiment (Figure 5) shows that for Muon, more than 5 iterations is overkill. This finding accords with standard practice, which implements Muon using just 5 iterations of Jordan’s method. Muon implemented with 5 iterations is taking steps in a different direction than Muon implemented with exact SVD, yet they both converge to the same loss. Why?
> > >
> > > We explore this question in Appendix H.1 of the latest revision. Our hypothesis is that the subspace corresponding to the smallest singular values of the momentum matrix does not contain much useful information. Therefore, the direction of the step we take in this subspace does not significantly impact the performance of the optimizer. In symbols, we divide the momentum matrix into two parts: $M = U_1 \Sigma_1 V_1^\top + U_2 \Sigma_2 V_2^\top$. The top singular values are in $\Sigma_1$, and the smaller singular values are in $\Sigma_2$. The exact Muon update, as computed by the SVD in Figure 5, is $U_1 V_1^\top + U_2 V_2^\top$; that is, the exact Muon update is akin to replacing each singular value with 1. We contend that only the top part is actually important. To test this hypothesis, we compare the exact Muon update to $U_1 V_1^\top$ and $U_1 V_1^\top - U_2 V_2^\top$ in Figure 10. This shows that—when we define “small” singular values to be those smaller than $\sigma_{\max}/10^3$, all three of these update rules perform well. (When “small” means smaller than $\sigma_{\max}/10^4$, they all perform identically.)
> > >
> > > Our hypothesis implies that, when approximating $\mathrm{polar}(M)$ for Muon, it is not important for *all* the singular values to converge to 1, only the “larger” ones. Therefore, Polar Express can perform well with just a few iterations, even if the small singular values have not converged. Figure 9 shows that, indeed, Polar Express does converge in just 5 or 6 iterations if we ignore singular values smaller than $\sigma_{\max} / 10^3$.
> > >
> > > We believe this hypothesis merits further study and that the experiments of Appendix H.1 are a promising step toward answering the scientific question raised by reviewer WacA.

---

### Official Review · Reviewer_kcTx · 2025-10-31

**Soundness:** 4
**Presentation:** 4
**Contribution:** 4
**Rating:** 10
**Confidence:** 4

**Summary:**

This paper seeks to improve the Muon optimizer which is being used in several recent results as an alternative to Adam/AdamW for training small-sized language models. In Muon, one subroutine is calculating the polar decomposition. This is the main task handled in this work. The standard solution involves running Newton-Schulz which works well but convergence can be slow. More recent heuristics to speed it up can work but may not converge to the correct solution. Polar Express proposes to fix by adapting the polynomial at each iteration (a minimax problem is solved). This gives good initial speed of the heuristics and guarantees on convergence. On the practical side, GPT-2 models are trained to a better validation loss. The paper is well written. The technical insights are very nice and the empirical results are convincing.

**Strengths:**

1.  The paper replaces recent heuristic strategies which can be fast but plateau out with a provably optimal algorithm. The initial progress is rapid and has similar convergence guarantees like classical methods. I found this result very interesting and new.

2. I was familiar with the Nakatsukasa/Freund paper, but the way in which the minimax paradigm is adapted here is surprising in a good way, giving Thm 3.1 and 3.3. I feel that the idea is quite novel and can be useful beyond the Muon use case.

3. The experimental results back up the main claims. Fig. 3 is sufficient validation that the idea in this paper deserves consideration.

4. What is also nice is that this is not just a algorithmic results paper. The authors discuss numerical instability by adjusting the polynomials, showing the adjustments needed for practical gains on modern GPUs. Same with precomputing coefficients etc. Excellent balance of technical findings with practicality.

**Weaknesses:**

1. Relatively minor. It will be good to show how much slower is Zolo-PD using newer QR implementations.
2. Also minor. Experiments are only on GPT-2 with Muon. If Muon is effective, why not expand the scope of experiments to other architectures and even finetuning?

**Questions:**

1. You may consider slightly expanding the scope of experiments. The paper is still a valuable contribution, but even if the idea is not equally effective in other cases, it will be good to point out where the gains are limited.

---

> ### Author Response · Authors · 2025-11-21
>
> Thank you so much for your review! We are honored that you liked the paper and especially by your comments on Theorems 3.1 and 3.3.
>
> Your suggestions are well taken. Our comments are as follows:
>
>
>
> ### Question 1: How much slower is Zolo-PD using newer QR.
>
> This is an interesting question. We expect that the GPU-friendliness of polynomial methods like PolarExpress will be hard to beat, especially for a low accuracy approximation. (See also our new Figure 5). However, we have not tried implementing Zolo-PD. You are right to point out that the implementation of QR is key. We use PyTorch for all our experiments, and its implementation of the QR decomposition may not be ideal; see [here](https://github.com/pytorch/pytorch/issues/22573#issuecomment-509657100).
>
> ### Question 2: Expand the scope of experiments to other architectures
>
> We are currently working on broadening the scope of our experiments to include Resnet and vision tasks. We will hopefully add some of these results to our paper before the end of the discussion period.

---

> > ### Author Response · Authors · 2025-12-03
> >
> > We have now added experiments with a different task (image classification) and a different architecture (ResNet) to Appendix H.3, as the reviewer suggested.

---

### Official Review · Reviewer_Xn5s · 2025-10-31

**Soundness:** 4
**Presentation:** 3
**Contribution:** 3
**Rating:** 8
**Confidence:** 4

**Summary:**

The paper proposes a better method for computing polar factor decomposition, practical for GPU implementation. Optimality under certain conditions is proven, and improved empirical performance for implementation into the Muon algorithm for synthetic problems and GPT model training.

**Strengths:**

An important and timely problem is considered with strong potential for practical impact.

A novel method is proposed and analyzed mathematically.

Reasonable empirical experiments illustrate and support the claims.

**Weaknesses:**

The set of practical ML problems can be diversified a bit, e.g., by considering a vision domain.

**Questions:**

Does the approximation quality depend on the dimension of the problem?

---

> ### Author Response · Authors · 2025-11-21
>
> Thank you for the kind review and your many helpful comments! We will address each point in turn.
>
> # Weaknesses:
> ### 1. “For GPT‑2 Large, 1B tokens is on the small side  (below Chinchilla optimal)”
> This point is well taken. Following your suggestion, we reran some of our GPT-2 experiments using 10 billion tokens, which is approximately Chinchilla optimal for GPT-2-Large and well beyond Chinchilla optimal for GPT-2-Small. We include the results in our revised paper. See Section 4.3, “Number of Training Tokens” and Figure 6, and see Appendix H, Figure 11. These experiments confirm that Polar Express consistently outperforms the baselines across all learning rates, though the effect size is somewhat smaller.
>
> ### 2. “The set of practical ML problems can be diversified a bit, e.g., by considering vision domain.”
> This is an excellent suggestion, and we are working on it now. We plan to include experiments with a small vision transformer in the next revision of this paper, hopefully before the end of the discussion period.
>
> # Minor issues, typos, and copy-editing
>
> We thank the reviewer for pointing out these issues, which we have now addressed. We offer two clarifications:
>
> ### 1. Weight decay
> We try all four combinations of GPT-Large vs. GPT-Small and weight decay = 0 vs. weight decay = 1. The presentation of these experiments has been reorganized for greater clarity. The full results are now presented in Figure 9 (no weight decay) and Figure 10 (yes weight decay) of Appendix H. Figures 1 and 4 in the main text now only present runs without weight decay. A reference to the experiments that use weight decay has been added to the main text in Section 4.3.
>
> ### 2. Attribution of You’s method
> The official citation is to a blog post coauthored by Cesista, You and Jordan. However, the idea is credited to Jiacheng You by the other authors. See for instance [Jiacheng’s six-step method](https://docs.modula.systems/algorithms/newton-schulz/#jiacheng-s-six-step).

---

> > ### Author Response · Authors · 2025-11-21
> >
> > We now address your six questions.
> >
> > # Questions:
> >
> > ### 1. “Many readers associate Kimi K2 with ~1T parameters”
> > You are right; Kimi K2 does have 1 trillion parameters. (We had mistakenly written “32 billion” because it is a mixture-of-experts model with 32 billion *active* parameters.) This has been corrected.
> >
> > ### 2. How do the Jordan and You heuristics fail to converge?
> > Jordan’s method applies the polynomial $3.4445x - 4.775x^3 + 2.0315 x^5$ at each iteration. No matter how many iterations are used, the error (measured in any norm on matrices) will not approach zero, because this polynomial does not have a fixed point at $x=1$. Instead, it has two fixed points at $x\approx 0.868$ and $x \approx 1.264$. Therefore, the singular values do not converge to 1, which is required for convergence, when repeatedly applying this polynomial. A minimal example is therefore the diagonal matrix with diagonal entries  $0.868$ and $1.264$. Another is the $1 \times 1$ matrix $\begin{bmatrix}0.75291\end{bmatrix}$, which seems to be part of a stable orbit of period 4.
> >
> > You’s method is only defined for six iterations (as described in the text) and therefore only achieves limited accuracy.
> >
> > In Section 4.1 we test Jordan’s and You’s methods on example matrices (see Figure 3). One can see that Jordan’s method fails to converge, since the error plateaus, and that You’s method is undefined after six iterations. However, let us emphasize that neither method was designed to converge to arbitrary precision. Rather, they were designed to quickly give a rough approximation of the polar factor.
> >
> > ### 3. Would you consider adding a non‑LM task (e.g., a vision model)
> > Yes, we are currently working on adding some vision experiments. We hope to be able to upload them before the end of the discussion period.
> >
> > ### 4. You emphasize minimizing the number of matrix–matrix products. Why not target total arithmetic cost?
> >
> > We agree that, in principle, minimizing the total arithmetic cost is reasonable. However, on a GPU, one must only rely on simple linear algebra routines such as matrix multiplications and additions to take full advantage of the hardware. Therefore, like the other methods for implementing Muon (Newton-Schulz, Jordan’s method, You’s method), our method uses just three types of operations: 1. matrix addition 2. matrix rescaling 3. matrix multiplication. Matrix multiplications are significantly more expensive than the others ($O(n^3)$ vs. $O(n^2)$), and therefore dominate the arithmetic cost. We do not believe it is possible to improve on the arithmetic cost of our method without using other kinds of operations besides these three.
> >
> >
> >
> > ### 5. Prior theory typically analyzes Muon with an exact polar/sign. For the camera‑ready, could you add a short discussion on how inexactness in Polar Express propagates in Muon?
> >
> > This is an excellent suggestion. We have added a comment after Theorem 3.3 citing two theoretical analyses of Muon with inexact polar factors (including SUMO) and noting that these results can be used to obtain guarantees for Muon + Polar Express.
> >
> > As for the effect of inexactness in Polar Express on Muon’s convergence, we added a new experiment studying this exact question. See Section 4.3, Figure 5. Our results show that if you run Polar Express for just 5 or 6 iterations, the inexactness has almost no effect on accuracy. They also show that using the SVD to compute the polar factor exactly is definitely overkill, since it dramatically increases runtime without improving the optimization quality.
> >
> > ### 6. Why restrict to odd monomials?
> >
> > For rectangular matrices, we cannot compute even monomials of the singular values without explicitly computing the SVD. Even for a square, but non-symmetric, matrix $M$, the matrix power $M^{2}$ does not correspond to squaring the singular values. We have now highlighted this difficulty in a footnote in Section 2. This is why we are restricted to using odd monomials.
> >
> > If we could compute even monomials of the singular values efficiently it would certainly help. In fact, if we could compute the zeroth power of the singular values, we would immediately have $M^0 = \mathrm{polar}(M)$.

---

> > > ### Author Response · Authors · 2025-12-03
> > >
> > > We have now added experiments with a different task (image classification) and a different architecture (ResNet) to Appendix H.3, as the reviewer suggested.

---

### Official Review · Reviewer_caWQ · 2025-11-01

**Soundness:** 3
**Presentation:** 3
**Contribution:** 3
**Rating:** 8
**Confidence:** 3

**Summary:**

This paper proposes a new algorithm for computing the matrix polar decomposition for the Muon optimizer used in training large neural networks. Unlike classical methods designed for high numerical accuracy, Polar Express is optimized for GPU efficiency and relatively low-precision computation (bfloat16). The authors derive the algorithm by solving a minimax optimization problem that guarantees optimal convergence in the worst case and demonstrate substantial empirical improvements over existing methods.

**Strengths:**

1. This paper introduces a theoretically grounded algorithm.
2. The method is practical, using GPU-efficient GEMMs and is numerically stable.
3. Experiments on GPT-2 with the Muon optimizer show consistent improvements in validation loss over existing methods.

**Weaknesses:**

1. It would be interesting to see how well the method works on models other than language models.
2. Lack of detailed runtime or throughput benchmarks. The experiments show the method works better than baselines under the same number of iterations, but it is unclear whether it is faster.

**Questions:**

see weakness section.

---

> ### Author Response · Authors · 2025-11-21
>
> Many thanks for your review!
>
> ### Q1. How well does the method perform on models other than language models?
>
> This is an excellent suggestion, and we are working on it now. We plan to include experiments with a small vision transformer in our final revision of this paper.
>
>
> ### Q2. The method works better than baselines under the same number of iterations, but is the method faster?
>
> The runtime of Polar Express is the same as that of the other baselines. This is because Polar Express applies a series of odd polynomials of degree 5, just like the Jordan, You, and Newton-Schulz baselines (see Algorithm 1, online stage). All that differs from previous methods are the coefficients of those polynomials.
>
> Furthermore, the figures in Appendix H show runtime vs validation loss when these methods are used within the Muon optimizer. These experiments confirm that Polar Express achieves a smaller validation loss in the same runtime.

---

> > ### Author Response · Authors · 2025-12-03
> >
> > We have now added experiments with a different task (image classification) and a different architecture (ResNet) to Appendix H.3, as the reviewer suggested.

---

### Author Response · Authors · 2025-11-21
**Revision on November 20**

We thank all the reviewers for their helpful feedback. We have uploaded a revised version of the paper that contains the following changes (highlighted in red in the PDF):
- Adding a sentence in the contributions (Section 1.3) noting that Polar Express has been incorporated into the modded-NanoGPT speedrun code base, which is a community lead, highly optimized implementation of LLM training. In modded-NanoGPT, Polar Express was used to set the current speedrun record, and continues to be the default method for computing the polar factor.
- Adding an experiment that compares implementing Muon using Polar Express with 5 iterations (our recommended method) to (a) Polar Express with a range of iterations from 2 to 30, and (b) the exact polar factor computed via an SVD. See Section 4.3 and Figure 5. This experiment shows that: less than 5 iterations performs worse, more than 6 iterations performs no better, and SVD performs no better while also being significantly slower.
- Adding an experiment that trains GPT-2-Large and GPT-2-Small on 10 billion tokens instead of 1 billion.
- Adding Figure 8 in Appendix H, which supplements Figure 3 by showing convergence in the Frobenius norm and cosine distance.
- Reorganizing the placement of our main experiments for greater legibility. Figures 1 and 4 present our experiments training GPT-2 without weight decay on 1 billion tokens, which are described in Section 4.2. Appendix H contains more results from those experiments (Figure 9). It also contains Figure 10, showing the same thing but with weight decay, and Figure 11, showing the same thing but with 10 billion training tokens. A new section, 4.3, describes these ablations in the main text.
- Adding small clarifications and correcting minor typos pointed out by the reviewers.

In addition to the new experiments described above, we are currently preparing experiments that use Muon to train a ResNet for image classification. We intend to have these online before the end of the discussion period.

---

> ### Author Response · Authors · 2025-12-03
>
> We have just uploaded a third version of our paper. It has the following changes compared to the second version:
>
> ### Appendix H.1
> We add several additional plots and experiments that address questions posed by reviewer WacA.
>
> ### Appendix H.3
> We add a suite of experiments on the CIFAR-10 and CIFAR-100 image classification datasets with ResNet and Vision Transformer architectures. This addresses reviewer comments that asked about datasets and architectures other than GPT language models. Overall, Polar Express performed among the best in terms of training and validation loss, consistent outperforming AdamW and SGD-M. However, differences between the various Muon methods were small, suggesting that the benefit of Polar Express for vision tasks over, e.g., simple Newton-Schultz iteration, requires further exploration.

---

### Meta-Review · Area_Chair_ADiC · 2026-01-07

**Summary:**

The paper introduces Polar Express, a GPU-friendly algorithm for computing low-accuracy polar decompositions (equivalently, matrix sign functions) optimized for use within the Muon optimizer for deep learning. Unlike classical numerical linear algebra methods that target high precision, Polar Express explicitly targets the regime relevant for modern training, that is, low precision (e.g., bfloat16), few iterations, and reliance solely on matrix–matrix multiplications to maximize GPU throughput.

The key technical contribution is an adaptive polynomial iteration derived from solving a minimax optimization problem at each step. The authors prove that this strategy is worst-case optimal within its polynomial class, yielding both fast early-iteration error reduction and asymptotically optimal convergence. This positions Polar Express as a principled alternative to existing heuristics (e.g., Newton–Schulz variants, Jordan/You methods), which may converge quickly initially but plateau or fail to converge.

Empirically, the method is evaluated primarily through integration into Muon for training GPT-2 models. Across a range of learning rates and training token budgets (from 1B to 10B tokens), Polar Express consistently matches or improves validation loss relative to prior Muon implementations, while avoiding the large runtime cost of exact SVD-based polar decomposition. Additional ablations examine iteration count, showing that roughly 5–6 iterations suffice in practice, and that exact polar computation via SVD does not yield better optimization outcomes despite much higher cost.

**Reviewer Concerns:**

Reviewers were generally very positive of the work. All concerns were considered relatively minor, but are nonetheless addressed by the authors.

- Numerical Experiments (raised by caWQ, Xn5s, kcTx, WacA): Multiple reviewers noted that the experimental evaluation was initially narrow, focusing almost exclusively on GPT-2 language models trained with Muon. This raised questions about whether the benefits of Polar Express generalize beyond this specific optimizer–model pairing (e.g., to vision models, other architectures, or other optimization contexts). The authors added CIFAR-10/100 experiments with ResNet and Vision Transformer models, as well as longer-horizon GPT-2 runs (10B tokens). While relegated to the appendix, I believe these additions help to address the concerns.

- Runtime (raised by caWQ, kcTx): Reviewers asked whether Polar Express is actually faster in wall-clock terms, as opposed to merely achieving lower loss for the same number of iterations. There was also interest in comparisons against alternative polar methods (e.g., Zolo-PD with modern QR). The authors argue that Polar Express has identical per-iteration cost to competing polynomial methods because it uses the same sequence of GEMMs, differing only in coefficients. They add runtime-vs-loss plots showing better loss at equal runtime within Muon, but do not implement or benchmark Zolo-PD. This is an adequate response in my view.

- Importance of Better Approximations (raised by WacA): Another concern was whether minimizing worst-case spectral norm error meaningfully aligns with what matters for Muon updates and whether better approximations actually translate to better training outcomes. In response, the authors added targeted ablations comparing different iteration counts and exact SVD-based Muon, showing that beyond ~5-6 iterations, increased accuracy does not improve training loss. They also included cosine similarity and Frobenius norm analyses and provided a theoretical argument linking spectral norm error to worst-case deviation in the Muon update. The response is thoughtful and effective.

- Overtrained Regimes (raised by WacA, Xn5s): Some reviewers were concerned that early gains might disappear in longer or overtrained regimes. The authors address this directly by adding 10B-token experiments, including regimes near or beyond Chinchilla optimality. These results confirm that the advantage of Polar Express persists but diminishes in effectiveness.

**Reviewer Scores:**

All reviewers expressed a good opinion of the work, ranging from a marginal accept (6) to a strong accept (10). Given their initial high score and limited room for improvements, it appears unlikely that Reviewers caWQ, Xn5s, and kcTx would have increased their score during the discussion period. Reviewer WacA expressed some skepticism in their initial review that was in line with their acknowledgement of lack of expertise in the area. Following up, they engaged with a response after the rebuttal. Judging by the quality of the author responses, I suspect that a score increase in this case seems likely, possibly to a score of 8, in line with the other reviewers.

---

### Decision · Program_Chairs · 2026-01-26

Accept (Oral)